



# Evaluated kinetic and photochemical data for atmospheric chemistry: Volume VII - Criegee intermediates

R. Anthony Cox[1], Markus Ammann[2], John N. Crowley[3], Hartmut Herrmann[4], Michael E. Jenkin[5], V. Faye McNeill[6], Abdelwahid Mellouki[7], Jürgen Troe[8], and Timothy J. Wallington[9]

[1] Centre for Atmospheric Science, Dept. of Chemistry, University of Cambridge, Lensfield Road, Cambridge CB2 1EP, UK
[2] Laboratory of Radiochemistry and Environmental Chemistry, OFLB 103, Paul Scherrer Institut, 5232 Villigen, Switzerland
[3] Max Planck Institute for Chemistry, Division of Atmospheric Chemistry, 55128 Mainz, Germany
[4] Leibniz Institute for Tropospheric Research (TROPOS), Atmospheric Chemistry Dept. (ACD), 04318 Leipzig, Germany
[5] Atmospheric Chemistry Services, Okehampton, Devon, EX20 4QB, UK
[6] Department of Chemical Engineering, Columbia University, New York, NY 10027, USA
[7] ICARE-CNRS, 1 C Av. de la Recherche Scientifique, 45071 Orléans CEDEX 2, France
[8] Institute of Physical Chemistry, University of Göttingen, Tammannstr. 6, 37077 Göttingen, Germany
[9] Ford Motor Company, Research and Advanced Engineering, Mail Drop RIC-2122, Dearborn, Michigan 48121-2053, USA

*Correspondence to*: M. E. Jenkin (atmos.chem@btinternet.com), J. N. Crowley (john.crowley@mpic.de) and R. A. Cox (rac26@cam.ac.uk)

**Abstract.** This article, the seventh in the series, presents kinetic and photochemical data sheets evaluated by the IUPAC Task Group on Atmospheric Chemical Kinetic Data Evaluation. It covers an extension of the gas phase and photochemical reactions related to Criegee intermediates previously published in ACP in 2006, and implemented on the IUPAC website up to 2020. The article consists of an introduction, description of laboratory measurements, discussion of rate coefficients for reactions of $O_3$ with alkenes producing Criegee intermediates, rate coefficients of unimolecular and bimolecular reactions and photochemical data for reactions of Criegee intermediates, and overview of the atmospheric chemistry of Criegee intermediates. Summary tables of the recommended kinetic and mechanistic parameters for the evaluated reactions are provided. Data sheets summarizing information upon which the recommendations are based are given in two files, provided as a Supplement to this article.

## 1 Introduction

Laboratory kinetic and mechanistic studies of the reactions of alkenes with ozone ($O_3$) have established that "Criegee intermediates" (CIs) produced from these reactions are potentially important oxidants in atmospheric chemistry (e.g. Calvert et al., 2000; Johnson and Marston, 2008; Taatjes et al., 2014). This followed the suggestion by Cox and Penkett (1971; 1972) that the rapid oxidation of $SO_2$ in the presence of reacting mixtures of $O_3$ and alkenes in air, was caused by production of a reactive intermediate, namely the peroxidic zwitterion, $R_1R_2C=O^+-O^-$, proposed by Rudolf Criegee, based on studies of the liquid phase ozonolysis of alkenes (e.g. Criegee et al., 1954).



This has led to extensive study of the mechanisms of $O_3$ + alkene reactions, and of the chemistry of the CIs formed. It is well established that the reaction proceeds by initial cyclo-addition of $O_3$ across the C=C double bond in an alkene to form an energy-rich primary ozonide (POZ), which rapidly decomposes to form either of two sets of CI and carbonyl compound, as shown in Fig. 1. The reaction is exothermic, leading to an excess of energy (200–250 kJ mol$^{-1}$) distributed between these

reaction products. Some of this excess energy is deposited as internal energy in the nascent CIs which can promote unimolecular decomposition, or which can be lost by collisional energy transfer to other gas molecules, leading to formation of stabilised Criegee intermediates (sCIs), which can themselves react with other atmospheric trace species. The impact of the ozonolysis reaction on atmospheric oxidation chemistry is therefore influenced by the relative importance of prompt CI decomposition vs. formation of sCI. The recognition of the important distinction between the chemically activated CI,

formed promptly in excited state, and the thermally equilibrated sCI has led to extensive experimental efforts to determine the yield of sCI ($Y$) formed from the ozonolysis of a variety of alkenes, as discussed further in Sect. 3. It is also well established that the decomposition of both CI and sCI leads to the formation of hydroxyl (HO) radicals and other radical products. Due to its important role in initiating the oxidation of organics (including alkenes), the formation of HO radicals has received particular attention, and this is discussed further in Sect. 4.

The mechanism in Fig. 1 shows that the CIs (and sCIs) formed from the ozonolysis of a simple alkene can each be formed as either of two stereo-isomers, with different orientations of the outer O atom relative to the substituent groups. This potentially has an important impact on the chemical pathways available, and their relative rates. The stereo-isomers have generally been distinguished using the terms *syn-* and *anti-*, to specify the orientation of the outer O atom relative to a particular substituent; although the use of the IUPAC *Z-* and *E-* nomenclature is becoming increasingly adopted (e.g.

Vereecken et al., 2017). As shown in Fig. 1, the ozonolysis of a simple alkene containing four different substituents therefore produces four distinct CIs, with this number being systematically reduced in symmetrical alkenes (because the products of the two POZ fragmentation pathways are the same); or in alkenes possessing two identical substituents on the same carbon atom (because the stereo-isomerism in the CI is removed). However, the number of different CI isomers can also be increased if the alkene contains alkenyl substituents, as is the case for the CIs formed from the ozonolysis of dienes such as

the $C_4$ species derived from isoprene, because of additional stereo-isomerism in the substituent group(s).

Based on current understanding of the mechanism of alkene ozonolysis (as illustrated in Fig. 1), the steady state concentration of a given stabilised Criegee intermediate, [sCI$_i$], maintained by a balance between production and loss, is described by Eq. (1):

$$[sCI_i] = \frac{\sum_j(Y_{ij} \times k_{1j} \times [\text{alkene}_j] \times [O_3])}{(k_{di} + J_i + k_{2i}[H_2O] + k_{3i}[(H_2O)_2] + k_{4i}[SO_2] + k_{5i}[NO_2] + \cdots + \cdots)} \qquad (1)$$

Here, $k_{1j}$ is the rate coefficient for the reaction of $O_3$ with alkene$_j$, and $Y_{ij}$ is the yield of sCI$_i$ from that reaction. The numerator of Eq. (1) therefore quantifies the source term for formation of sCI$_i$ from all relevant alkenes, and the denominator





quantifies the sum of the rates of the unimolecular and individual bimolecular loss processes for $sCI_i$, with the example contributors to the summation being based on the processes shown in Fig. 1. This illustrates that knowledge of the yields, rate coefficients and products for the component reactions is important for quantitative description of the chemical pathways controlling the atmospheric chemistry and impact of the given $sCI_i$. It also shows the importance of establishing how these

parameters vary from one sCI to another, e.g. the structural-dependence of the rate coefficients for their unimolecular and bimolecular reactions.

Although rate constants have been determined accurately for a large number of $O_3$ + alkene reactions, using both direct and relative rate techniques, all kinetic data reported for sCI reactions prior to 2012 were based on indirect relative rate techniques. Many of these data were previously evaluated by the IUPAC Task Group on Atmospheric Chemical Kinetic

Data Evaluation (Atkinson et al., 2006). Since 2012, many new rate coefficients for sCI reactions have been reported using direct kinetic studies, providing a wealth of data for the elementary reaction kinetics and spectroscopy of sCI reactions. The current evaluation therefore addresses these reactions, extending substantially the scope of our former evaluation published in ACP in 2006. This includes a major extension of the scope of the evaluation to include rate coefficients of elementary reactions of selected sCIs, which have provided a better understanding of the atmospheric impact of sCI chemistry.

In this review we summarize the results of this evaluation activity, presenting in turn the recommended kinetic data for the key reactions in the above mechanism, using data for those species which are representative of the chemistry of the terrestrial atmosphere. The rate coefficients for $O_3$ + alkene initiation reactions are presented and discussed in Sect. 2, with reference to a series of detailed data sheets which are provided in Supplement A. Information on the sCI and HO radical yields from the ozone + alkene reactions is presented in Sects. 3 and 4, with additional discussion once again provided in the corresponding

data sheets in Supplement A. The data sheets therefore each include summary information on the reported kinetics studies of the given reaction, and provide an overview of reported mechanistic information and product yields where available.

The spectroscopy and kinetics recommendations for the sCI reactions are presented and discussed in Sects. 5 and 6. These include data for bimolecular and unimolecular reactions of selected sCIs of particular atmospheric relevance for which direct kinetic data have been reported, namely $CH_2OO$, *Z*- and *E*- $CH_3CHOO$ and $(CH_3)_2COO$; and we also provide some

discussion of the $C_4$ intermediates formed from isoprene (see Fig. 2 for sCI structures). These are predicted to be among the most important sCIs in tropospheric chemistry (Vereecken et al., 2017), and can also act as a systematic set of template species for representing the fates of some larger and more complex sCIs. Detailed data sheets for the sCI reactions are provided in Supplement B, providing supporting summary information and discussion. Finally, the recommended kinetics parameters are used to evaluate the relative importance of the different fates of the sCIs under representative atmospheric

conditions in Sect. 7, and an overview of the impact of Criegee intermediates in atmospheric oxidation chemistry is given in Sect. 8.





## 2 Rate coefficients of $O_3$ + alkene reactions

The present evaluation considers the reactions of $O_3$ with 31 alkenes, including small ($C_1$ to $C_4$) alkenes, isoprene, monoterpenes and sesquiterpenes. This represents a substantial increase on those considered in our previous evaluation (Atkinson et al., 2006), which were limited to ethene, propene, isoprene and α-pinene. The reactions are listed in Table 1,

along with the associated recommended rate coefficients. A detailed data sheet for each reaction is also provided in Supplement A. As discussed in detail previously (e.g. Calvert et al., 2000; 2015), the data indicate that the rate coefficients are highly sensitive to alkene structure, and depend on the degree of alkyl substitution of the unsaturated bond(s), on steric effects and on ring-strain effects in cyclic compounds. The lower tropospheric lifetimes of the alkenes, with respect to reaction with 20 ppb $O_3$, therefore cover several orders of magnitude, ranging from as short as 2-3 minutes for reactive

species such as α-terpinene, β-caryophyllene and α-humulene, to about seven weeks or longer for camphene and longifolene. For the simple alkenes, the lifetimes range from about 30 minutes for the fully-substituted 2,3-dimethylbut-2-ene to about two weeks for ethene. With the exception of the least reactive compounds, removal by ozonolysis is expected to make a contribution for all the evaluated alkenes under lower tropospheric conditions, and is generally the dominant fate for those with rate coefficients in excess of about $10^{-15}$ $cm^3$ molecule$^{-1}$ s$^{-1}$.

## 15  3 sCI yields from $O_3$ + alkene reactions

As described in Sect. 1 and Fig. 1, the chemically-activated CIs formed from the $O_3$ + alkene reactions may either decompose promptly, or lose energy by collisions with other molecules to form stabilised Criegee intermediates (sCIs). The sCIs have the potential to undergo reactions with other atmospheric trace gases leading to their oxidation and formation of characteristic products. Thus, it is important to quantify the yield ($Y$) of each sCI from each relevant precursor alkene if the

impact of alkene ozonolysis and Criegee chemistry on oxidation processes is to be correctly represented in atmospheric mechanisms, or to allow the local steady-state concentration of the sCIs to be estimated by Eq. (1).

There have been extensive experimental efforts to determine $Y$ for a variety of $O_3$ + alkene systems. The yields have generally been expressed as a fraction of the molar amount of $O_3$ reacted, and have been determined by reaction of the sCIs formed with an appropriate added scavenger reagent (e.g. $SO_2$, $H_2O$, HCHO, HCOOH, $CF_3C(O)CF_3$). The yield is

determined either from quantitative analysis of a characteristic product of the sCI + scavenger reaction, or through measurement of the loss of the scavenger. The reported values of $Y$ are therefore indirect measurements which have generally quantified the total yield of sCIs formed in a given alkene + $O_3$ system, with little or no information on the contributions of the component sCI species for asymmetric alkenes being reported. Table 2 gives a summary of the recommended values of $Y$ for the evaluated $O_3$ + alkene reactions at 298 K and 1 bar, with additional details provided in the

corresponding reaction data sheets in Supplement A. Values for selected other alkenes (*trans*-dec-5-ene and cyclohexene) are also given to help illustrate structural variations in $Y$, as discussed further below.



As indicated in Sect. 1, the stabilisation of the promptly formed chemically-activated CIs requires loss of internal energy through collisional energy transfer to other gas molecules. As a result, studies of a number of $O_3$ + alkene systems have shown that $Y$ depends on pressure (e.g. Hatekayama et al., 1986; Drodz and Donahue, 2011; Hakala and Donahue, 2016; 2018; Campos-Pineda and Zhang, 2017), but with significant residual values at zero pressure. This indicates that a limiting yield of sCIs (i.e. with internal energy below the threshold required for decomposition or isomerization) is typically formed directly from decomposition of the primary ozonide, POZ (although this is not illustrated in Fig. 1 for simplicity), with additional sCI formation resulting from collisional stabilisation of the chemically-activated CIs.

The value of $Y$ is also expected to vary systematically with alkene structure, and the reported data display some logical structural trends that are consistent with theoretical treatments. These can be rationalized in terms of the nascent internal energy in the Criegee intermediate, and how this is influenced by partitioning of the excess energy into either translational modes (i.e. recoil), internal energy of the carbonyl co-product, or into non-reactive vibrational or rotational modes within the Criegee intermediate (e.g. Choung et al., 2004; Drozd et al., 2011). The value of $Y$ is therefore expected to increase with alkene size along a homologous series, as has been confirmed for a set of *trans-* symmetric alkenes between $C_4$ and $C_{14}$ by Hakala and Donahue (2018). As a result, the values at 298 K and 1 bar for large acyclic alkenes (e.g. $Y = 1.0$ for *trans*-dec-5-ene and *trans*-tetradec-7-ene) tend to be larger than those for the smaller acyclic alkenes, as listed in Table 2. Similarly, those for large cycloalkenes with endocyclic double bonds (e.g. $Y > 0.6$ for the $C_{15}$ β-caryophyllene) tend to be larger than those for smaller species such as the $C_{10}$ α-pinene ($Y = 0.18$) and the $C_6$ cyclohexene ($Y < 0.05$). It is also clear that the values for cycloalkenes with endocyclic double bonds are systematically lower than those for similarly sized acyclic alkenes (or cycloalkenes with exocylic double bonds). This can be rationalised in terms of the excess energy being confined within a single product possessing both Criegee and carbonyl functionalities for the cycloalkenes with endocyclic double bonds; whereas it can be dissipated into translational modes, and the internal energy of the carbonyl co-product, for acyclic alkenes (and for cycloalkenes with exocylic double bonds).

It is also recognized that the indirect yield measurements are often subject to significant uncertainties, and variability between different measurement methods (e.g. see Hakala and Donahue, 2016), such that systematic trends can be masked. Another important consideration is that exceptionally rapid unimolecular decomposition and isomerization reactions are predicted to be available for some sCIs, e.g. some of the $C_4$ species formed from $O_3$ + isoprene (Vereecken et al., 2017), as is discussed further in Sects. 4 and 7. In these cases it is possible that the bimolecular reactions with scavengers are unable to compete with the fast unimolecular processes, and consequently the reported yields for sCI formation may be underestimated.

## 4 HO radical yields from $O_3$ + alkene reactions

It has been established for several decades that the reactions of $O_3$ with alkenes lead to the formation of HO radicals (e.g. Finlayson et al., 1972; Donahue et al., 1998), and considerable attention has been given to quantifying HO radical yields for





many $O_3$ + alkene systems. In most of the reported studies, the yields have been determined indirectly by reaction of the HO formed with an appropriate added scavenger reagent (e.g. cyclohexane, 1,3,5-trimethylbenzene, butan-2-ol). The yield is then determined either from quantitative analysis of the yield of a characteristic product of the HO + scavenger reaction, or through measurement of the loss of the scavenger. However, direct detection methods (particularly LIF) have also been used

to quantify HO yields in a number of studies (e.g. Donahue et al., 1998; Siese et al., 2000; Kroll et al., 2001a; 2001b; Malkin et al., 2010; Alam et al., 2013), providing unequivocal identification of HO as a product of $O_3$ + alkene reactions, and a means of confirming the validity and interpretation of the indirect methods (e.g. Malkin et al., 2010). Table 3 gives a summary of the recommended HO radical yields for the evaluated $O_3$ + alkene reactions at 298 K and 1 bar, with additional details provided in the corresponding reaction data sheets in Supplement A.

The formation of HO radicals can result from both prompt unimolecular decomposition of chemically-activated CIs, and from decomposition of thermally-equilibrated sCIs over longer timescales (e.g. as demonstrated in the time-resolved pressure dependence measurements of Kroll et al., 2001c). The most important mechanism forming HO is generally accepted to proceed by a 1,4 H shift isomerization to an excited vinyl hydroperoxide intermediate, which decomposes to form HO and a vinoxy or β-oxo alkyl radical; and this has been characterised for a variety of Criegee intermediates in

theoretical studies (e.g. Vereecken et al., 2017). Using *Z*-$CH_3CHOO$ as an example, the mechanism proceeds as shown below:



(R6)

First proposed for $(CH_3)_2COO$ by Niki et al. (1987), this mechanism is therefore potentially available for all di-substituted and *Z*- mono-substituted Criegee intermediates that possess a β-hydrogen atom, but is unavailable for $CH_2OO$ and *E*- mono-substituted Criegee intermediates (e.g. *E*-$CH_3CHOO$), where the outer O atom of the CI moiety is not directed towards an

organic group. In the cases where reaction (R6) is available, it is believed to be the dominant decomposition route for small (e.g. methyl- and ethyl-substituted) Criegee intermediates. The reported variation of HO yields with structure for simple small alkenes (e.g. as shown in Table 3) can therefore be broadly rationalized in terms of the combination of Criegee intermediates formed, and whether or not reaction (R6) is available; and this has provided the basis of simple structure-activity relationships (SARs) for HO yields from $O_3$ + alkene reactions (e.g. Rickard et al., 1999). Thus, those for fully

methyl-substituted alkenes (e.g. 0.93 for 2,3-dimethyl-but-2-ene) tend to be systematically higher than those for partially methyl-substituted alkenes (e.g. 0.33 for *cis*-but-2-ene and 0.60 for *trans*-but-2-ene), which in turn are higher than that for the unsubstituted ethene (0.17).

Another key route involves initial rearrangement (1,3 ring-closure) to form a dioxirane intermediate (see Fig. 1). This provides an alternative unimolecular decomposition pathway for $CH_2OO$ and *E*- mono-substituted Criegee intermediates





(e.g. $E$-CH$_3$CHOO), and is also calculated to be competitive for some $Z$- mono-substituted Criegee intermediates possessing oxygenated substituents (Vereecken et al., 2017). In the former case, it is likely to be significant only for the chemically-activated CI, [CH$_2$OO]*, because unimolecular loss of the thermally-equilibrated sCI, CH$_2$OO, is observed and calculated to be slow (see Sect. 7). The dioxirane intermediate isomerizes to form "hot" formic acid, [HC(O)OH]*, which can fragment

via a number of pathways:

$$[HC(O)OH]^* \rightarrow HCO + HO \text{ (or } H + CO + HO) \tag{R7a}$$

$$\rightarrow CO + H_2O \tag{R7b}$$

$$\rightarrow CO_2 + H_2 \tag{R7c}$$

$$\rightarrow CO_2 + H + H \tag{R7d}$$

Reaction (R7a) can therefore account for the small HO yield (0.17) resulting from the O$_3$ + ethene reaction (Table 3), and the set of product channels can also rationalize the observed formation of HO$_2$ (from the reactions of O$_2$ with H and HCO), CO and CO$_2$ (see data sheet OX_VOC5 in Supplement A). In the cases of $E$- mono-substituted Criegee intermediates (e.g. $E$-CH$_3$CHOO), isomerization via a dioxirane intermediate is again expected to be significant for chemically-activated CIs, and is also calculated to occur for thermally-equilibrated sCIs (e.g. Vereecken et al., 2017). For sCIs, however, it is in

competition with some particularly rapid bimolecular reactions under atmospheric conditions (see Sects. 7 and 8), and may therefore be of limited importance. In the case of the chemically activated [$E$-CH$_3$CHOO]*, the dioxirane intermediate isomerizes to form "hot" acetic acid, [CH$_3$C(O)OH]*, and the following pathways can rationalize the observed formation of HO$_2$, CO$_2$, CH$_2$=CO (ketene), CH$_4$ and CH$_3$OH from the reactions of O$_3$ with propene and $cis$- and $trans$-but-2-ene (see data sheets OX_VOC6 , OX_VOC17 and OX_VOC18  in Supplement A1).

$$[CH_3C(O)OH]^* \rightarrow CH_2=CO + H_2O \tag{R8a}$$

$$\rightarrow CO + CH_3OH \tag{R8b}$$

$$\rightarrow CO_2 + CH_4 \tag{R8c}$$

$$\rightarrow CO_2 + H + CH_3 \tag{R8d}$$

In principal, formation of HO (and CH$_3$CO, or CH$_3$ and CO) may also occur, by a route analogous to reaction (R7a). The

possible contribution of the corresponding channel more generally for $E$- mono-substituted CIs is poorly characterised, although it is generally accepted to be only a minor source of HO compared with the 1,4 H shift isomerization route for the $Z$- conformers. The dioxirane route has also been reported to lead to the formation of stabilised acid, ester or lactone products, particularly in larger O$_3$ + akene systems (e.g. Hakola et al., 1994; Griesbaum et al., 1998; Winterhalter et al., 2009; Nguyen et al., 2009a,b).

A number of other unimolecular reactions are available for Criegee intermediates possessing larger organic substituents, particularly those that are unsaturated (e.g. see Vereecken et al., 2017). For example, very rapid 1,5 ring-closure reactions are expected to dominate for $Z$-$\alpha,\beta$-unsaturated sCIs, such as $Z$-(CH=CH$_2$)(CH$_3$)COO and $Z$-(C(CH$_3$)=CH$_2$)CHOO formed from O$_3$ + isoprene, e.g. for $Z$-(CH=CH$_2$)(CH$_3$)COO:



(R9)

In the cases of carbonyl-substituted sCIs, such as those formed from cycloalkenes with endocyclic double bonds, the potential for (bicyclic) ring-closure to form intramolecular secondary ozonides is well established, e.g.:

(R10)

These reactions have been characterised in a number of theoretical studies (e.g. Chuong et al., 2004; Nguyen et al., 2009b; Mackenzie-Rae et al., 2016; Vereecken et al., 2017; Long et al., 2019), with experimental evidence for their formation also

reported (e.g. Winterhalter et al., 2009; Vibenholt et al., 2009; Beck et al., 2011). In practice, however, these reactions are only observed (and calculated) to be significant for larger systems (e.g. sesquiterpene ozonolysis), where the Criegee intermediates are formed significantly stabilised, and the ring-closure reaction does not result in prohibitive ring-strain. Where these criteria are met, they are predicted to be rapid reactions that can compete with, or dominate over, other decomposition routes; and this in one factor contributing to the low HO yields reported for some sesquiterpenes (e.g. β-

caryophyllene and α-humulene).

## 5 Structure and spectroscopy of Criegee intermediates

Assessment of the photolysis rates and product channels for sCIs requires quantitative data for the absorption cross sections and quantum yields for individual sCI species at atmospheric, actinic wavelengths, mainly in the UV and visible. The development of methods for creating specific sCIs in defined concentrations has allowed experimental investigation of their

spectroscopy and structure, e.g. see the review by Osborn and Taatjes (2015). Moreover, advances in theoretical methods have also provided insight into the spectra and structure of sCIs, and quantum calculations have given further details of reaction mechanisms and product channels of sCI photolysis (e.g. Samanta et al., 2014).

The spectroscopic studies of sCIs have led to the recognition that they have a singlet ground electronic state whose dominant configuration is that of a zwitterion, and this is reflected in the large dipole moment of these species (Chhantyal-Pun et al.,

2017a). The observed spectra of the $C_1 - C_3$ sCIs exhibit strong and broad absorptions centred in the near UV, with maximum cross sections of the order of $10^{-17}$ cm$^2$ molecule$^{-1}$. These features, and their detailed rovibrational structures, are consistent with B(1A') ← X (1A') transitions, i.e. intense $\pi^* \leftarrow \pi$ transitions analogous to the familiar UV spectrum of $O_3$ in



the Hartley/Huggins bands. Photodissociation of sCIs from this excitation yields $O(^1D)$ and a carbonyl compound with a quantum yield of unity.

## 5.1 Conformers

As described in Sect. 1 and Fig. 1, Criegee intermediates with dissimilar substituents can exist as two possible conformers,
denoted $E$- and $Z$-, which differ in the orientation of the outer O atom relative to the substituent groups. The simplest examples are $E$- and $Z$-$CH_3CHOO$ (acetaldehyde oxide, see Fig. 2), for which conformer dependence has been demonstrated experimentally and theoretically in its spectra and its reaction rates and pathways. Calculations place $Z$-$CH_3CHOO$ about 15 kJ mol$^{-1}$ lower in energy than $E$-$CH_3CHOO$ (Kuwata et al., 2010), reflecting the zwitterionic character of the Criegee intermediate structure. Calculated energies of the $E$- and $Z$- conformers of $CH_3CHOO$ are consistent with the spectral shift of
$\lambda_{max}(Z$-$) < \lambda_{max}(E$-$)$ shown in Fig. 3. This has enabled conformer-specific reactions to be investigated, using direct observation of the kinetics and products of the two conformers. The barrier to interconversion is substantial (~160 kJ mol$^{-1}$), and consequently $E$- and $Z$-$CH_3CHOO$ act as distinct chemical species at atmospheric temperatures. The absence of rotation is an important indicator of the zwitterionic character of the intermediate, as originally proposed by Criegee et al. (1954).

## 5.2 UV spectra of stabilised Criegee intermediates

In experimental studies of UV-visible spectra, the series of $C_1 - C_3$ sCIs have been formed by photolysis of the corresponding di-iodoalkane (*via* C-I bond fission), followed by the reaction of the iodoalkyl radical with $O_2$, e.g. in the case of $CH_2OO$:

$CH_2I_2 + h\upsilon \rightarrow CH_2I + I$                                                             (R11)

$CH_2I + O_2 \rightarrow CH_2OO + I$                                                          (R12)

Absolute cross sections at specific wavelengths have been derived by monitoring the laser UV-induced depletion of the sCIs, monitored, for example, by mass spectrometry or time resolved UV-absorption spectroscopy. The experimental data reveal some discrepancies regarding the shapes, structure, and intensities of the $B(^1A') \leftarrow X(^1A')$ spectra determined using transient absorption spectroscopy, compared with laser-induced depletion techniques, determined under molecular beam conditions. This discrepancy has been attributed to the much lower temperatures reached in the molecular beams, compared to
measurements at ambient temperature, but lack of detailed data on the temperature dependence of the cross sections precludes firm conclusions to be drawn. The evaluation of the spectral data, and detailed discussion of the reported studies, is given for the $C_1 - C_3$ sCIs in the data sheets in Supplement Sect. B5. The recommendations for the maximum absorption cross sections are given in Table 4, and the spectra are presented in Fig. 3. Using these data, representative lower tropospheric photolysis removal rates in the range 0.4−1.4 s$^{-1}$ can be calculated for a solar zenith angle of 30° at the surface,
based on the actinic flux estimates of Madronich, presented by Finlayson-Pitts and Pitts (2000). This indicates that loss by





photolysis is likely only a minor or negligible loss process for sCIs in the lower atmosphere, compared with their collective removal by the thermal reactions discussed in the following section.

Novel methods for the production of the $C_4$ isoprene-derived sCIs have also been reported (Barber et al., 2018; Vansco et al., 2019), with $Z$- and $E$-$(CH=CH_2)(CH_3)COO$ formed from the photolysis of 1,3-di-iodobut-2-ene and $Z$- and $E$-$(C(CH_3)=CH_2)CHOO$ from the photolysis of 1,3-di-iodo-2-methylprop-1-ene, both in the presence of $O_2$. This has allowed characterization of the UV-visible spectra of the unsaturated $C_4$ sCIs (Vansco et al., 2018; 2019), which are reported to be broader and shifted to longer wavelengths compared with those of the simple $C_1 - C_3$ sCIs as a result of the conjugation of the vinyl and carbonyl oxide groups (see detailed discussion in the data sheets in Supplement Sect. B5).

## 6 Rate coefficients for thermal reactions of sCIs

### 6.1 Measurements of absolute rate coefficients for reactions of sCIs

In recent years, numerous direct studies of the elementary reaction kinetics of sCIs have been reported. This has been made possible by two developments. First, the discovery of a novel fast photochemical source of sCIs, from the reactions of iodo-alkyl radicals with $O_2$, has allowed generation of specific sCIs, following the photolysis of the corresponding di-iodoalkane; as shown above in Sect. 5 for the example of $CH_2OO$ (reactions (R11) and (R12)). Second, direct time-resolved detection and measurements of sCI concentrations have been achieved using spectroscopic methods involving both tunable vacuum UV multiplexed photoionization mass spectrometry (MPIMS), and UV or IR absorption.

The first breakthrough in these developments for direct studies of sCI kinetics came from the work of Taatjes and co-workers, who used MPIMS to monitor the time-resolved decay of $CH_2OO$ in the presence of bimolecular reaction partners such as $SO_2$, NO and $NO_2$ (Welz et al., 2012; Taatjes et al., 2012). Subsequently it was shown that this technique for detection and production could be equally well applied to kinetics studies of the larger Criegee intermediates (e.g. Taatjes et al., 2013; Chhantyal-Pun et al., 2017b), so that structural effects on the basic oxidation rates and mechanisms could be explored directly.

The tunable light sources needed for MPIMS are not readily available for conventional laboratory rate constant measurements. The discovery of the strong absorption spectrum of $CH_2OO$ in the mid UV (Sheps, 2013) offered a second, more flexible and sensitive (but less specific) detection method for following sCI kinetics, which has the advantage of the ability to monitor sCI kinetics at up to 1 bar pressure, appropriate for lower atmospheric conditions. The UV absorption method is also applicable to the larger Criegee intermediates, produced from the same source chemistry (e.g. Sheps et al., 2014; Huang et al., 2015), and can provide kinetic/spectroscopic distinction of the $Z$- and $E$- conformers, where applicable.

The lower atmosphere contains many alkenes from both natural and man-made sources, which react with $O_3$ to form Criegee intermediates with a wide variety of structures. Earlier work on ozone-alkene reactions gave little clue on the structural dependence of sCI reactivity with trace gases such as $SO_2$, $H_2O$, $NO_2$ and organics, or of their unimolecular decomposition





rates. Direct kinetic studies have provided new information on the reaction rate constants and mechanisms of $C_1 - C_3$ sCIs formed from ozonolysis of simple alkenes, although direct experimental determinations of rate coefficients have not yet been reported for other, more complex species ($\geq C_4$), derived for example from biogenic alkenes such as isoprene and terpenes. However, as indicated in Sect. 5.2, methods for their production and spectroscopic characterization are emerging (e.g. Barber

et al., 2018; Vansco et al., 2018; 2019), providing a basis for direct kinetics studies of more structurally complex sCIs.

## 6.2 Evaluation of rate coefficients for bimolecular reactions

As noted in Sect. 1, most of the information on the kinetics of sCI reactions up to 2006 was based on data obtained using indirect relative rate techniques. These were evaluated by the IUPAC Task Group on Atmospheric Chemical Kinetic Data Evaluation and published in ACP in 2006 (Atkinson et al., 2006). Since 2012, the direct kinetics studies described above

have provided a wealth of new data on the elementary reaction kinetics and spectroscopy of sCIs. This has stimulated further competitive rate studies using static and slow-flow experiments in chambers to generate sCIs from $O_3$ + alkene reactions under atmospheric pressure and temperature conditions. In this evaluation, recommended rate coefficients are generally based on the results of direct kinetic studies of the sCIs, derived from di-iodoalkane precursors as described above (Sects. 5.2 and 6.1). However, the results of relative rate studies are also used to assess information on the kinetics, and to check for

consistency of the kinetic data for $C_1 - C_3$ sCIs, when produced by alkene ozonolysis. Table 5 provides a summary of the preferred values of bimolecular reaction rate coefficients, with additional details given in the corresponding reaction data sheets in Supplement B.

There are currently no direct kinetics determinations for reactions of the $C_4$ sCIs derived from isoprene. In these cases, the recommendations are either inferred from those for the simpler $C_2$ and $C_3$ species, or adopted from reported theoretical

studies (Vereecken et al., 2017; Chhantyal-Pun et al., 2017a). The performance of the ensemble of bimolecular reactions with $SO_2$, $H_2O$ and $(H_2O)_2$ (and unimolecular decomposition reactions) has been checked for consistency, using the results of published chamber and slow-flow experiments (Sipilä et al., 2014; Newland et al., 2015; Nguyen et al., 2016), as described in detail in data sheet CGI_21 (Supplement B). The results provide some confidence in these recommendations for use in practical applications, although the data also support some tolerance in the parameter values applied. Measurements of

speciated sCI yields, and direct kinetics studies of the rate coefficients and product channels for the reactions of the $C_4$ sCI isomers, are therefore required to allow these recommendations to be confirmed or refined.

## 6.3 Evaluation of rate coefficients for unimolecular decomposition

Table 6 shows a summary of preferred values of unimolecular decomposition rate coefficients, $k_d$, given in the corresponding data sheets in Supplement B. The evaluations are based on both consideration of direct time-resolved measurements, and

those reported in relative rate experiments, where $k_d$ is determined relative to the loss rate of sCI via a well-defined



competing bimolecular reaction. In the former case, $k_d$ is determined from observation of the decay kinetics of the Criegee intermediates themselves (or of a marker species) in the absence of a second reagent, or by extrapolation of the observed first order removal rate vs. reagent concentration plots to zero. Although unimolecular decomposition can make a major, or the dominant, contribution to such limiting first-order removal rates, other processes also need to be taken into account (e.g. wall

loss, reaction with impurity gases or diffusive loss from the monitoring probe area), as is usually discussed in the individual studies. In some cases, therefore, direct kinetics studies can only provide upper limit estimates of unimolecular decomposition rates for sCIs, particularly when the decomposition rate is slow.  Relative rate determinations can also be influenced by background loss processes for the sCI (e.g. reaction with impurity gases or products), and reported rate coefficient ratios need to be placed on an absolute basis using the rate coefficient for the competing, reference bimolecular

reaction, which itself has an associated uncertainty. In the present evaluations, the reference (bimolecular) rate coefficients are all based on the preferred values given in Table 5.

As indicated above in Sect. 6.2, the recommendations for the unimolecular decomposition of $C_4$ sCIs derived from isoprene ozonolysis are adopted from the theoretical study of Vereecken et al. (2017). These recommendations have been assessed, along with the recommended bimolecular rate coefficients for reactions with $SO_2$, $H_2O$ and $(H_2O)_2$, using the results of

published chamber and slow-flow experiments (Sipilä et al., 2014; Newland et al., 2015; Nguyen et al., 2016), as described in detail in data sheet CGI_21 (Supplement B).

## 7 Overall reactivity conclusions – comparison of experiment and theory

In addition to the progressive increase in the availability of experimental data, there have been substantial advances in the theoretical treatment of structure and reaction kinetics of sCIs in the gas phase (e.g. Olzmann et al., 1997; Zhang et al., 2002;

Anglada et al., 2002; 2011; Ryzhkov and Ariya, 2004; Kuwata et al., 2010; Vereecken et al., 2012; 2017; Liu et al., 2014; Anglada and Solé, 2016; Vereecken and Nguyen, 2017; Chhantyal-Pun et al., 2017a). Theoretical studies provided particular guidance prior to the advances in sCI production and detection methods for direct kinetics measurements, as described above. However, the body of experimental information now available for a series of sCIs allows the results of theoretical studies to be validated, and for the methods to be refined, optimised and extended.

In this section, the recommended rate coefficients for unimolecular decomposition and bimolecular reactions of the $C_1 – C_3$ sCIs with $H_2O$ and $(H_2O)_2$ at 298 K and atmospheric pressure are compared with those derived from theoretical calculations (see Table 7). The calculated values are based on those reported in the comprehensive study of Vereecken et al. (2017), which presented theory-based structure-activity relationships (SARs) for 98 atmospherically relevant classes of sCI. Those for unimolecular decomposition reactions were derived from consideration of 14 reaction types (e.g. including the 1,4 H shift

isomerization, 1,3 ring-closure and 1,5 ring-closure reactions discussed and illustrated in Sect. 4) for a benchmark series of sCIs containing key substituents, comprising a set of about 170 calculated rate coefficients. Those for the bimolecular reactions with $H_2O$ and $(H_2O)_2$ were based on fitting theory-derived reactivity trends to a set of literature data, which





included rate coefficients available at the time for the same set of $C_1 - C_3$ sCIs considered in the present evaluation. As a result, the absolute scaling of the theory-based SAR values cannot be considered to be entirely independent of our experimentally-based evaluations for the $H_2O$ and $(H_2O)_2$ reactions, although comparison of the (considerable) reactivity variation across the series of sCIs is valid.

The comparisons shown in Table 7 demonstrate that the theory-based SAR rate coefficients reported by Vereecken et al. (2017) show a good level of consistency with our recommended rate coefficients for the reactions of the $C_1 - C_3$ sCIs. Both sets of parameters display a similar structural dependence across the series of sCIs. The rate coefficients for the bimolecular reactions with $H_2O$ and $(H_2O)_2$ agree to within about a factor of about four (which is well within the combined uncertainties), where direct comparison is possible (i.e. for $CH_2OO$ and $E$-$CH_3CHOO$); and where only an upper limit recommendation is

possible in the present work (i.e. for $Z$-$CH_3CHOO$ and $(CH_3)_2COO$), the SAR rate coefficient is fully compatible with that recommendation. This indicates a consistent structure-reactivity variation across the series, with systematically higher reactivities for the reactions of $CH_2OO$ and $E$-$CH_3CHOO$ with both $H_2O$ and $(H_2O)_2$.

The unimolecular decomposition parameters recommended in the present study for $E$- and $Z$-$CH_3CHOO$ and $(CH_3)_2COO$ are in very good agreement with the theory-based SAR values; being consistent with dominant 1,3 ring-closure to form a

dioxirane intermediate for $E$-$CH_3CHOO$, and dominant 1,4 H shift isomerization to form vinyl hydroperoxide intermediates for $Z$-$CH_3CHOO$ and $(CH_3)_2COO$. In the case of $CH_2OO$, however, the approximate rate coefficient recommended in the present work, based on extrapolation of higher temperature experimental data (see data sheet CGI_12), is two orders of magnitude lower than the theory-based SAR value, although both values are sufficiently slow for unimolecular decomposition to be uncompetitive under tropospheric conditions.

Also shown in Table 7 are representative lower tropospheric first-order loss rates ($k_I$) (at 298 K and atmospheric pressure) calculated for the same series of reactions using both sets of rate parameters; and for the bimolecular reactions with $SO_2$, based on the rate coefficients recommended here. The representative rates for the bimolecular reactions assume a mid-range relative humidity of 40 % (corresponding to $[H_2O] \approx 3 \times 10^{17}$ molecule cm$^{-3}$ and $[(H_2O)_2] \approx 2 \times 10^{14}$ molecule cm$^{-3}$), and an $SO_2$ concentration of $2.5 \times 10^{10}$ molecule cm$^{-3}$ (about 1 ppbv), broadly typical of urban background air. Although these

concentrations are only representative, the associated values of $k_I$ nevertheless give a clear indication that unimolecular decomposition is the dominant loss route for $Z$-$CH_3CHOO$ and $(CH_3)_2COO$, consistent of a widespread role for 1,4 H shift isomerization for $Z$- monoalkyl-substituted and dialkyl-substituted sCIs possessing a β-hydrogen atom. In contrast, bimolecular reactions with $H_2O$ and/or $(H_2O)_2$ are the dominant loss routes for $CH_2OO$ and $E$-$CH_3CHOO$, and likely most other $E$- monoalkyl-substituted sCIs. The values of $k_I$ also give an indication that loss due to reaction with $SO_2$ generally

only makes a minor contribution to sCI removal, away from the immediate vicinity of $SO_2$ sources, such that their ambient concentration is mainly controlled by either decomposition or reaction with $H_2O$ and/or $(H_2O)_2$.

The match between the experimentally-based recommendations presented here, and those derived from the theory-based SARs for this set of sCIs, gives some confidence that the SAR rate coefficients of Vereecken et al. (2017) provide a very reasonable basis for representing the structural dependence of the kinetic parameters for unimolecular decomposition and





bimolecular reactions with $H_2O$ and $(H_2O)_2$. In view of this, our corresponding recommendations for the more complex $C_4$ isoprene-derived species (*E*- and *Z*-(CH=CH$_2$)(CH$_3$)COO and *E*- and *Z*-(C(CH$_3$)=CH$_2$)CHOO) are currently adopted from Vereecken et al. (2017), as evaluated and discussed in detail in data sheet CGI_21 (Supplement B). The 298 K rate coefficients and representative first order loss rates ($k_I$) are also shown for these $C_4$ sCIs in Table 7. The values of $k_I$ clearly

demonstrate that bimolecular reactions cannot compete with the very rapid unimolecular decomposition of *Z*-(CH=CH$_2$)(CH$_3$)COO and *Z*-(C(CH$_3$)=CH$_2$)CHOO (via 1,5 ring-closure) under atmospheric conditions (and indeed most reported experimental conditions); and that decomposition of *Z*-(CH=CH$_2$)(CH$_3$)COO (via 1,4 H shift isomerization) is also likely to be its major loss route, by virtue of its slow bimolecular reactions with $H_2O$ and $(H_2O)_2$. In the case of *E*-(C(CH$_3$)=CH$_2$)CHOO, however, loss via unimolecular decomposition and bimolecular reactions with $H_2O$ and $(H_2O)_2$ are

predicted to occur at comparable rates, and experimental confirmation of the rate coefficients would be of particular value for this species.

## 8 Impact of Criegee intermediates in atmospheric oxidation chemistry

The kinetics and mechanistic information for sCI reactions recommended in the present evaluation provides the basis for representing the associated impact of alkene ozonolysis in atmospheric chemical mechanisms. The significance of sCIs as

atmospheric oxidants can be discussed in terms of Eq. (1), which defines the local balance between production and loss of sCIs, and hence their steady-state concentrations. As described in more detail in Supplement C, the parameters recommended in this evaluation (supplemented by data from other sources) have been used to calculate surface production rates, loss rates and steady-state concentrations of a series of sCIs for average ambient conditions representative of rural background, suburban background and urban kerbside (urban traffic) locations in the south-east UK in both winter and summer. The

calculations make use of measured or inferred concentrations of a series of $C_1 - C_6$ alkenes, isoprene, α-pinene, limonene, $O_3$, $NO_2$, $SO_2$ and HC(O)OH; in conjunction with $H_2O$ and $(H_2O)_2$ concentrations based on modelled temperature, and relative humidity data typical of the region. In this section, the key results are summarized and placed in context by comparison with reported results calculated for other locations, and in global modelling studies.

### 8.1 sCI production rates

The formation of a series of 28 $C_1 - C_{10}$ sCIs from the speciation of 19 precursor alkenes was represented in the present calculations (see Supplement C for full details). The core set of $C_1 - C_4$ sCIs specifically considered in the present evaluation collectively makes an important contribution to the total production rate at each of the three locations; the respective winter and summer contributions lying in the ranges $88 - 91$ % and $42 - 85$% (see Table 8). The speciation of this core set of sCIs is also presented in Fig. 6 for the example of the rural background location (Chilbolton Observatory). The formation of *E*- and

*Z*-CH$_3$CHOO is most significant, because they are formed from propene and all the alk-2-enes in the applied speciation. They





are also favoured because reaction with $O_3$ is a major (and sometimes the dominant) removal route for alk-2-enes (and other internal alkenes), because of their particularly rapid reactions with $O_3$ (see Table 1). $CH_2OO$ also makes a notable contribution to the totals, because it is formed from all the alk-1-enes and isoprene. The higher alkyl-substituted sCIs make systematically lower contributions, primarily because their precursor alkenes have systematically lower abundances (see

Table 8).

As also shown in Table 8, sCIs formed from biogenic hydrocarbons logically make an increased and important collective contribution under the summer conditions. This is particularly the case at the suburban background location (London Eltham), which has a mixture of trees and other vegetation in close proximity to the measurement site. The monoterpene ($\alpha$-pinene and limonene) derived sCIs are calculated to be particularly significant, because ozonolysis generally makes a major

contribution to the removal of endocyclic alkenes, by virtue of their particularly rapid reactions with $O_3$ (see Table 1). In contrast, the reaction of $O_3$ with isoprene is comparatively slow, its dominant removal reaction being with HO radicals. Therefore, the production rates of the isoprene-derived sCIs are approaching an order of magnitude lower than those of the monoterpene-derived sCIs under the conditions represented here. However, isoprene-derived sCIs have been shown to make more important contributions globally, particularly in specific regions such as the Amazonian rain forest (e.g. Vereecken et

al., 2017; Khan et al., 2018), as a result of the dominant contribution of isoprene to global biogenic VOC emissions.

## 8.2 sCI loss rates

Based on the information presented and discussed in Sects. 6 and 7, sCI removal by unimolecular decomposition and bimolecular reactions with $H_2O$, $(H_2O)_2$, $NO_2$, $SO_2$ and $HC(O)OH$ was taken into account. Table 8 and Fig. 6 present information on the speciated and total first-order loss rates of the sCIs, and the contributions made by the series of removal

reactions considered. Consistent with the analysis presented in Sect. 7, the results confirm that the major loss routes for most of the sCIs are either thermal decomposition, or reaction with $(H_2O)_2$, supplemented by reaction with $H_2O$. As a result, these reaction classes dominate total sCI removal under all conditions, with reaction with $(H_2O)_2$ and $H_2O$ being slightly more important in the winter, and thermal decomposition being slightly more important in the summer. As also indicated in Sect. 7, thermal decomposition tends to dominate the removal of $Z$- mono-substituted and di-substituted sCIs, with reaction with

$(H_2O)_2$ and $H_2O$ dominating the removal of $CH_2OO$ and $E$- mono-substituted sCIs.

The total first-order loss rates for the individual sCIs lie in approximate range 20 s$^{-1}$ to 20,000 s$^{-1}$ for the full series of considered conditions (see Supplement C). Those toward the low end of the range generally correspond to sCIs for which the dominant removal route is 1,4 H atom migration, occurring at only a modest rate (e.g. as in the cases of $Z$-$CH_3CHOO$ and $E$-$(CH=CH_2)(CH_3)COO$), particularly under winter conditions; and for which the reactions with $(H_2O)_2$ and $H_2O$ are very slow.

Those at the high end of the range generally correspond to $E$- monosubstituted sCIs (e.g. $E$-$CH_3CHOO$) for which the dominant removal reactions with $(H_2O)_2$ and $H_2O$ are very fast. The associated loss rates for selected sCIs in the former category are sufficiently slow for removal by reaction with $HC(O)OH$ and $SO_2$ (and to a lesser extent, reaction with $NO_2$) to



make a notable contribution (e.g. $E$-(CH=CH$_2$)(CH$_3$)COO) in Fig. 6). In collective terms, however, these classes of reaction each makes only a small (< 4 %) contribution to total sCI removal under the series of conditions considered here, as shown in Table 8.

## 8.3 Steady-state concentrations and speciation

The total first-order sCI loss rates indicated above correspond to individual sCI lifetimes lying in the range 50 µs to 50 ms, confirming that calculation of their concentrations using the steady-state approximation, described by Eq. (1), is valid. The resultant calculated steady-state concentrations of the core set of $C_1 - C_4$ sCIs for the rural background conditions are shown in Fig. 4, with the total sCI concentrations for the three scenarios given in Table 8. The totals calculated for the rural background conditions (389 molecule cm$^{-3}$ and 381 molecule cm$^{-3}$ for winter and summer, respectively) are broadly

consistent with the low annual average values simulated for the UK in the global modelling calculations of Vereecken et al. (2017); and the concentrations calculated for the series of locations and conditions (up to 1100 molecule cm$^{-3}$) are comparable with those reported by Khan et al. (2018), based on similar UK calculations to those reported here.

The distributions of sCIs are generally dominated by a limited number of individual species, and show similarities to those reported elsewhere for locations with significant anthropogenic VOC emissions (e.g. Vereecken et al., 2017). $Z$-CH$_3$CHOO

is the most abundant sCI for all the considered scenarios, accounting for 76-79 % of the totals for winter conditions, and 24-77 % of the totals for summer conditions. This results from the combination of its high production rate (see Fig. 4) and its relatively slow removal rate (see Fig. 5), as discussed above. Its lowest contribution occurs for summer conditions at the suburban location, when the sCIs derived from α-pinene and limonene (and to a lesser extent, isoprene) are collectively dominant (67 %), as discussed in more detail in Supplement C. Given the relatively low biogenic hydrocarbon emission rates

in the UK, this result for a suburban location in London site gives a strong indication that biogenic hydrocarbon derived sCIs will dominate the global concentrations, as clearly demonstrated in the modelling studies presented by Vereecken et al. (2017) and Khan et al. (2018). Those studies reported important and widespread global contributions from sCIs derived from both isoprene and monoterpenes, although they report substantially different sCI concentrations. In the work of Vereecken et al. (2017), the rapid unimolecular decomposition rates calculated for many of the of sCIs (as also adopted in the present

work) strongly suppress the simulated concentrations compared with those reported by Khan et al. (2018). The resultant annual average sCI concentration at the surface maximizes at $7 \times 10^3$ molecule cm$^{-3}$ over the Amazon basin, but is generally less than $2 \times 10^3$ molecule cm$^{-3}$ over most of the globe (Vereecken et al., 2017). This further emphasizes the need for direct kinetics studies of a structurally diverse series of isoprene and terpene-derived sCIs, to help validate and refine the rate coefficients calculated in theoretical studies.





## 8.3 Oxidation of SO₂ and organic acids

The speciated sCI distributions have also been used to calculate the associated SO$_2$ oxidation rates. As shown in Table 8, the total oxidation rates are calculated to be between 0.006 % h$^{-1}$ and 0.031 % h$^{-1}$. They broadly follow the simulated trend in total sCI concentrations, but also reflect that the rate coefficient values for individual species span almost an order of magnitude. These oxidation rates can be compared with a reference SO$_2$ oxidation rate of about 0.3 % h$^{-1}$ for reaction with HO radicals at a concentration of 10$^6$ molecule cm$^{-3}$. This comparison is therefore consistent with the < 10 % annual average contribution to gas phase SO$_2$ oxidation for the UK, reported in the global modelling calculations of Vereecken et al. (2017) and Khan et al. (2018). The more widespread potential role of biogenic hydrocarbon derived sCIs in global SO$_2$ oxidation has also been considered in those modelling studies. Although the results possess some similarities, in terms of the relative regional variation, their role is much more limited in the Vereecken et al. (2017) calculations, because of the high calculated decomposition rates applied to many of the sCIs, and the resultant suppression of sCI concentrations commented on above. Nevertheless, annual average contributions of up to about 70 % were still simulated for the terrestrial equatorial belt.

The largest bimolecular rate coefficients for sCI reactions that have been measured experimentally are those for reactions with organic acids such as HC(O)OH, CH$_3$C(O)OH and CF$_3$C(O)OH (see Table 5). HC(O)OH and CH$_3$C(O)OH are present in the troposphere in significant concentrations (e.g. Andreae et al., 1988; Millet et al., 2015; Bannan et al., 2017) due in part to their formation in the photochemical oxidation or ozonolysis of many VOCs, from both manmade and natural sources. As discussed above, and shown more widely in the calculation of Vereecken et al. (2017), the reactions with organic acids can make small but significant contributions to sCI removal in some regions. The total sCI concentrations calculated here suggest associated oxidation rates of (0.04 – 0.11) % h$^{-1}$ for HC(O)OH, and similar rates for CH$_3$C(O)OH. This can be compared with reference oxidation rates of about 0.16 % h$^{-1}$ and 0.25 % h$^{-1}$, for reaction of HC(O)OH and CH$_3$C(O)OH with HO radicals at a concentration of 10$^6$ molecule cm$^{-3}$. This indicates that reaction with sCIs makes an important contribution to the oxidation of these acids under the conditions considered here, with oxidation rates comparable to those via HO reaction calculated for equatorial regions in the global modelling study of Vereecken et al. (2017). It is noted that the reactions of larger sCIs and organic acids (e.g. derived from the ozonolysis of monoterpenes and sesquiterpenes) potentially forms highly oxidized, low-volatility products (hydroperoxyl-esters), which may play a role in secondary organic aerosol (SOA) formation (e.g. Tobias and Ziemann, 2001; Chhantyal-Pun et al., 2018).

As shown in Table 5, the reactions of sCIs with CF$_3$C(O)OH are particularly rapid. The total sCI concentrations calculated here suggest an associated oxidation rate of (0.1 – 0.3) % h$^{-1}$ for CF$_3$C(O)OH, compared with 0.25 % h$^{-1}$ for its reaction with HO radicals at a concentration of 10$^6$ molecule cm$^{-3}$. This demonstrates the potential importance of sCIs as gas phase oxidants for CF$_3$C(O)OH (and other perfluoro-carboxylic acids, C$_n$F$_{2n+1}$C(O)OH) over land masses. However, it is noted that the reaction of the resultant hydroperoxyl-fluoroester products with HO radicals probably reforms the perfluoro-carboxylic acids on a timescale of 1−2 days (Taatjes et al., 2019).





Current understanding of the atmospheric chemistry of sCIs therefore supports the original hypothesis of Cox and Penkett (1971; 1972) regarding their potential importance as atmospheric oxidants; identified from chamber measurements of $SO_2$ oxidation associated with alkene ozonolysis, and the observed effect of relative humidity on the oxidation rates. Considerable progress has since been made in the understanding the kinetics and mechanisms of alkene ozonolysis, resulting from both experimental and theoretical studies, with particular advances since the pioneering work of Taatjes and co-workers less than a decade ago (e.g. Welz et al., 2012; Taatjes et al., 2013) in the detection of sCIs and direct measurements of the kinetics of their reactions. However, significant uncertainties remain in some aspects of mechanistic understanding, including measurements of the yields of sCIs and their speciation in asymmetric alkene systems. The current evaluation has focused primarily on those sCIs for which direct kinetics measurements are available (i.e. $CH_2OO$, $E$-$CH_3CHOO$, $Z$-$CH_3CHOO$ and $(CH_3)_2COO$), with some consideration also given to the $C_4$ intermediates formed from isoprene. Whilst these represent an important subset of atmospheric sCIs, it is recognized that an enormous variety of sCIs are generated, with particularly important global contributions from those generated from the ozonolysis of monoterpenes and sesquiterpenes. The concurrent progress in the theoretical treatment of the structure and reaction kinetics of sCIs in the gas phase has allowed the development of theory-based structure activity relationships (SARs) (Vereecken et al., 2017), which provide a basis for representing the reactions of structurally complex sCIs in atmospheric mechanisms. As a result, there is a need for direct kinetics studies of a structurally diverse series of isoprene and terpene-derived sCIs, to help validate and refine the rate coefficients calculated in theoretical studies.

*Author contributions*. All authors defined the scope of the work. RAC, MEJ and TJW developed and drafted the data sheets and manuscript.  All authors reviewed, refined, and revised the manuscript and data sheets.

*Competing interests*. The authors declare that they have no conflict of interest.

## Acknowledgements

The Chairman and members of the Task Group wish to express their appreciation to I.U.P.A.C. for the financial help that facilitated the preparation of this evaluation. We also acknowledge financial support from the following organisations: Department of Chemistry, University of Cambridge, the Swiss National Science Foundation, the Office Fédéral de l'Education et de la Science, the Centre National de la Recherche Scientifique-Institut National des Sciences de l'Univers (CNRS-INSU), Orleans University, and Observatoire des Sciences de l'Univers en région Centre (OSUC). We thank Cathy Boone for developing and maintaining the website.



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





**Figure 1: Mechanism of the reaction of ozone with alkenes, showing formation of Criegee intermediates (CIs). The general types of reaction available for stabilised Criegee intermediates (sCIs) are also illustrated for one example. The substituents $R_1$ to $R_4$ can be either H atoms or organic groups, although the illustrated sCI/CI vinyl hydroperoxide route is unavailable if $R_2 =$ H. Note that in the case of endocyclic C=C bonds in cycloalkenes, the initially formed carbonyl and CI moieties are substituents of the same organic product.**


$CH_2OO$  $Z\text{-}CH_3CHOO$  $E\text{-}CH_3CHOO$  $(CH_3)_2COO$

$Z\text{-}(CH{=}CH_2)(CH_3)COO$  $E\text{-}(CH{=}CH_2)(CH_3)COO$  $Z\text{-}(C(CH_3){=}CH_2)CHOO$  $E\text{-}(C(CH_3){=}CH_2)CHOO$

**Figure 2: Structures of the stabilised Criegee intermediates considered in the present study, and the nomenclature assigned. In the cases of the di-substituted (isoprene-derived) C₄ intermediates, the Z- and E- notations specify the orientation of the first named substituent (which has the higher Cahn-Ingold-Prelog priority) relative to the CI moiety. The displayed rotamers of the C₄ intermediates are calculated to be in near equilibrium under atmospheric conditions (Vereecken et al., 2017), and are assumed to act as a single species in each case.**

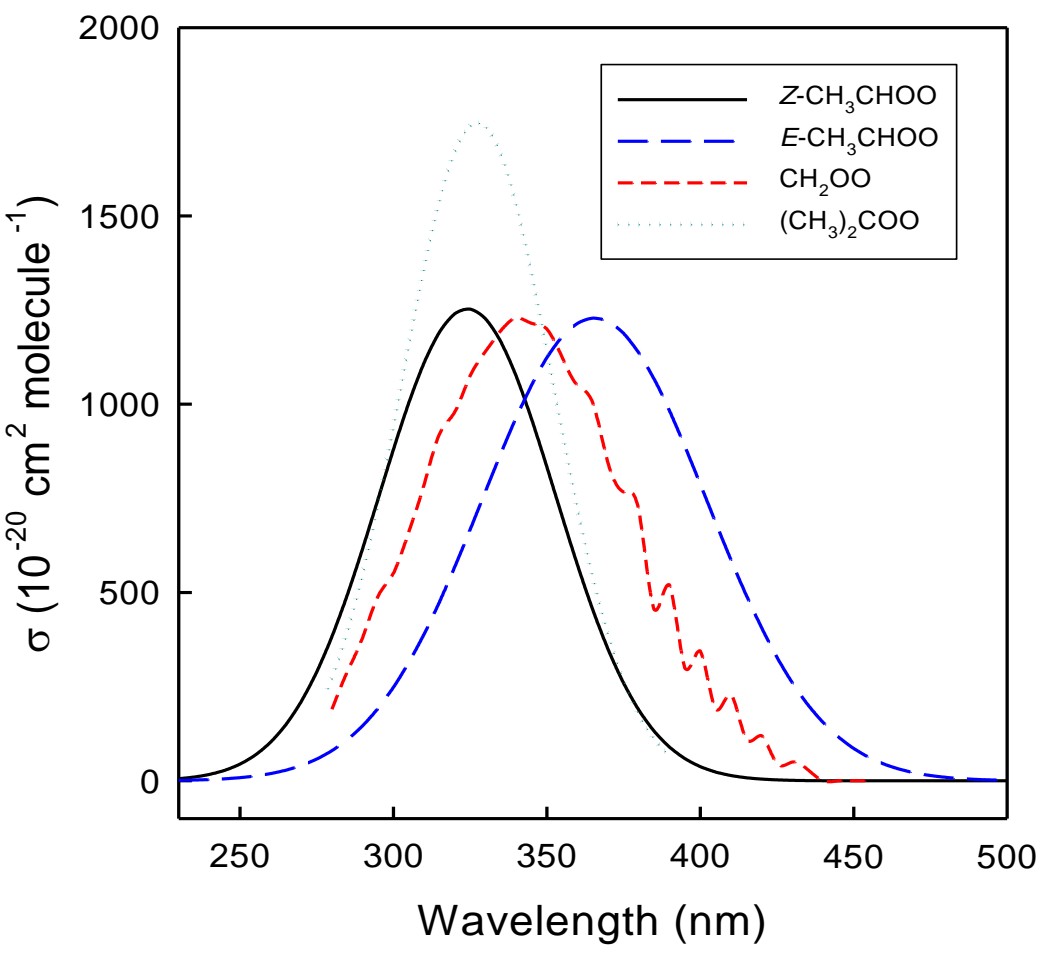

**Figure 3: Recommended spectra for CH₂OO, Z-CH₃CHOO, E-CH₃CHOO and (CH₃)₂COO.**



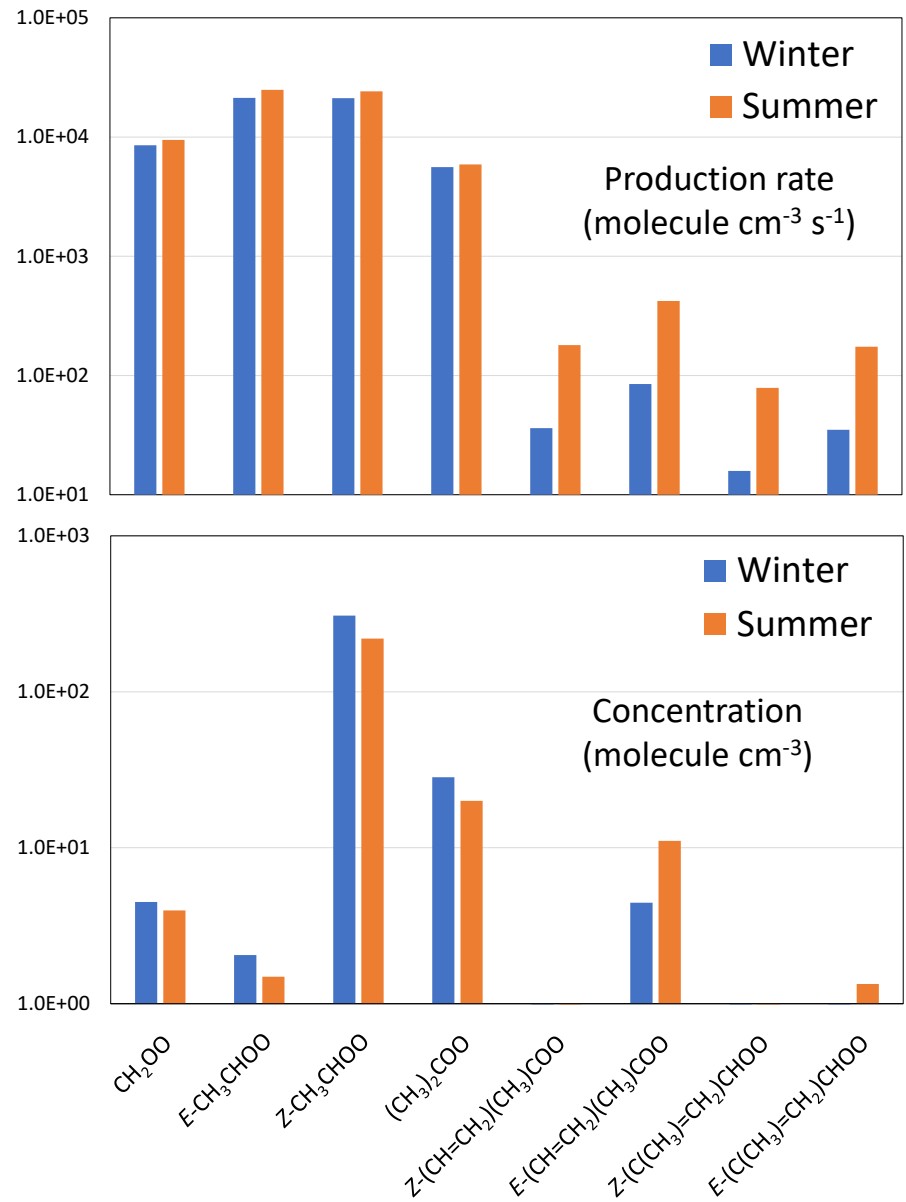

**Figure 4: Production rates (upper panel) and steady-state concentrations (lower panel) for the core set of $C_1$ – $C_4$ sCIs for representative rural background conditions in the south-east UK. Note that the information is presented on log scales, with cut-offs of 10 molecule $cm^{-3}$ $s^{-1}$ and 1 molecule $cm^{-3}$, respectively. The total sCI production rates and concentrations are given in Table 8. Results for an extended series of 28 sCIs, and for suburban and urban traffic conditions, are presented in Supplement C.**







**Figure 5:** Contributions of the unimolecular and bimolecular removal routes for the core set of $C_1 – C_4$ sCIs under winter conditions (upper panel) and summer conditions (lower panel) for representative rural background conditions in the south-east UK. The data label shows the total removal rate (in $s^{-1}$) for the given sCI. Results for an extended series of sCIs, and for suburban and urban traffic conditions, are presented in Supplement C.





**Table 1. Summary of recommended rate coefficients for reactions of O₃ with alkenes**

| Reaction ID | Alkene | $k_{298}$ cm³ molecule⁻¹ s⁻¹ | $\Delta \log k_{298}$ | $k(T)$ cm³ molecule⁻¹ s⁻¹ | T range | $\Delta(E/R)$ K |
|---|---|---|---|---|---|---|
| *Small alkene reactions – based on data sheets in Supplement A Sect. A1* | | | | | | |
| Ox_VOC5 | ethene | $1.55 \times 10^{-18}$ | $\pm 0.08$ | $6.82 \times 10^{-15} \exp(-2500/T)$ | 180-360 | $\pm 100$ |
| Ox_VOC6 | propene | $1.05 \times 10^{-17}$ | $\pm 0.15$ | $5.77 \times 10^{-15} \exp(-1880/T)$ | 230-370 | $\pm 100$ |
| Ox_VOC16 | but-1-ene | $1.0 \times 10^{-17}$ | $\pm 0.08$ | $3.55 \times 10^{-15} \exp(-1750/T)$ | 220-370 | $\pm 200$ |
| Ox_VOC17 | *cis*-but-2-ene | $1.3 \times 10^{-16}$ | $\pm 0.05$ | $3.37 \times 10^{-15} \exp(-970/T)$ | 220-370 | $\pm 200$ |
| Ox_VOC18 | *trans*-but-2-ene | $2.0 \times 10^{-16}$ | $\pm 0.1$ | $7.0 \times 10^{-15} \exp(-1060/T)$ | 220-370 | $\pm 200$ |
| Ox_VOC15 | 2-methylpropene | $1.15 \times 10^{-17}$ | $\pm 0.05$ | $2.92 \times 10^{-15} \exp(-1650/T)$ | 220-370 | $\pm 200$ |
| Ox_VOC41 | 2,3-dimethylbut-2-ene | $1.1 \times 10^{-15}$ | $\pm 0.08$ | $3.0 \times 10^{-15} \exp(-300/T)$ | 220-370 | $\pm 200$ |
| Ox_VOC7 | isoprene | $1.28 \times 10^{-17}$ | $\pm 0.08$ | $1.05 \times 10^{-14} \exp(-2000/T)$ | 240-360 | $\pm 200$ |
| *Monoterpene reactions – based on data sheets in Supplement A Sect. A2* | | | | | | |
| Ox_VOC8 | α-pinene | $9.6 \times 10^{-17}$ | $\pm 0.15$ | $8.22 \times 10^{-16} \exp(-640/T)$ | 240-370 | $\pm 300$ |
| Ox_VOC19 | β-pinene | $1.9 \times 10^{-17}$ | $\pm 0.25$ | $1.39 \times 10^{-15} \exp(-1280/T)$ | 290-370 | $\pm 300$ |
| Ox_VOC20 | limonene | $2.2 \times 10^{-16}$ | $\pm 0.1$ | $2.91 \times 10^{-15} \exp(-770/T)$ | 290-370 | $\pm 300$ |
| Ox_VOC21 | camphene | $5.0 \times 10^{-19}$ | $\pm 0.3$ | $9.0 \times 10^{-18} \exp(-860/T)$ | 285-315 | $\pm 500$ |
| Ox_VOC22 | 2-carene | $2.4 \times 10^{-16}$ | $\pm 0.2$ | | | |
| Ox_VOC23 | 3-carene | $4.9 \times 10^{-17}$ | $\pm 0.2$ | | | |
| Ox_VOC24 | β-myrcene | $4.7 \times 10^{-16}$ | $\pm 0.2$ | $2.69 \times 10^{-15} \exp(-520/T)$ | 290-320 | $\pm 300$ |
| Ox_VOC25 | β-ocimene | $5.1 \times 10^{-16}$ | $\pm 0.2$ | $4.15 \times 10^{-15} \exp(-625/T)$ | 290-320 | $\pm 300$ |
| Ox_VOC26 | α-phellandrene | $2.9 \times 10^{-15}$ | $\pm 0.2$ | | | |
| Ox_VOC27 | β-phellandrene | $5.2 \times 10^{-17}$ | $\pm 0.3$ | | | |
| Ox_VOC28 | sabinene | $8.3 \times 10^{-17}$ | $\pm 0.15$ | | | |
| Ox_VOC29 | α-terpinene | $1.9 \times 10^{-14}$ | $\pm 0.2$ | | | |
| Ox_VOC30 | γ-terpinene | $1.6 \times 10^{-16}$ | $\pm 0.3$ | | | |
| Ox_VOC31 | terpinolene | $1.6 \times 10^{-15}$ | $\pm 0.15$ | | | |
| *Sesquiterpene reactions – based on data sheets in Supplement A Sect. A3* | | | | | | |
| Ox_VOC32 | β-caryophyllene | $1.2 \times 10^{-14}$ | $\pm 0.15$ | | | |
| Ox_VOC33 | α-cedrene | no recommendation (see data sheet) | | | | |
| Ox_VOC34 | α-copaene | $1.5 \times 10^{-16}$ | $\pm 0.3$ | | | |
| Ox_VOC35 | α-farnesene | $5.9 \times 10^{-16}$ | $\pm 0.3$ | $3.5 \times 10^{-12} \exp(-2590/T)$ | 290-320 | $\pm 500$ |
| Ox_VOC36 | β-farnesene | $5.6 \times 10^{-16}$ | $\pm 0.25$ | $1.5 \times 10^{-12} \exp(-2350/T)$ | 290-320 | $\pm 500$ |
| Ox_VOC37 | α-humulene | $1.2 \times 10^{-14}$ | $\pm 0.15$ | | | |
| Ox_VOC38 | isolongifolene | $1.0 \times 10^{-17}$ | $\pm 0.3$ | | | |
| Ox_VOC39 | longifolene | $< 5 \times 10^{-19}$ | | | | |
| Ox_VOC40 | valencene | no recommendation (see data sheet) | | | | |





**Table 2. Summary of recommended total sCI yields ($Y$) from $O_3$ + alkene reactions at 298 K and 1 bar**

| Reaction ID [a] | alkene | $Y$ | comments |
|---|---|---|---|
| *Small alkene reactions* | | | |
| Ox_VOC5 | ethene | $0.42 \pm 0.10$ | (b) |
| Ox_VOC6 | propene | $0.25 \pm 0.10$ | (c) |
| Ox_VOC16 | but-1-ene | $\sim 0.27$ | (d) |
| Ox_VOC17 | *cis*-but-2-ene | $0.38 \pm 0.10$ | (e) |
| Ox_VOC18 | *trans*-but-2-ene | $0.43 \pm 0.10$ | (f) |
| Ox_VOC15 | 2-methylpropene | $\sim 0.17$ | (g) |
| Ox_VOC41 | 2,3-dimethylbut-2-ene | $0.38 \pm 0.10$ | (h) |
| Ox_VOC7 | isoprene | $0.65 \pm 0.10$ | (i) |
| *Monoterpene and sesquiterpene reactions* | | | |
| Ox_VOC8 | α-pinene | $0.18 \pm 0.05$ | (j) |
| Ox_VOC19 | β-pinene | $0.5 \pm 0.1$ | (k) |
| Ox_VOC20 | limonene | $0.32 \pm 0.14$ | (l) |
| Ox_VOC21 | camphene | $\sim 0.31$ | (d) |
| Ox_VOC32 | β-caryophyllene | $> 0.6$ | (m) |
| *Selected other reactions* | | | |
| - | cyclohexene | $< 0.05$ | (n) |
| - | *trans*-dec-5-ene | $1.0$ | (o) |
| - | *trans*-tetradec-7-ene | $1.0$ | (p) |

**Comments**

[a] See corresponding data sheets in Supplement A for further information; [b] Based on Su et al. (1980), Kan et al. (1981), Hatekayama et al. (1984; 1986), Horie and Moortgat (1991), Neeb et al. (1996; 1998), Horie et al. (1999), Hasson et al. (2001a), Alam et al. (2011), Newland et al. (2015a); [c] Based on Hatekayama et al. (1984) and Horie and Moortgat (1991); [d] Based on Hasson et al. (2001b); [e] Based on Newland et al. (2015a); [f] Based on Berndt et al. (2014), Newland et al. (2015a) and Hakala and Donahue (2018). Pressure dependence measurements suggest $Y$ falls to ~25 % at 50 Torr (Hakala and Donahue, 2018); [g] Based on Hatekayama et al. (1986); [h] Based on Berndt et al. (2014), Newland et al. (2015a) and Hakala and Donahue (2016). Pressure dependence measurements suggest $Y$ falls to 12-15 % at zero pressure (e.g. Hakala and Donahue, 2016; Campos-Pineda and Zhang, 2017); [i] Based on Sipilä et al. (2014), Newland et al. (2015b) and Nguyen et al. (2016), as also discussed further in data sheet CGI_21 (Supplement B); [j] Based on Drozd and Donahue (2011) and Sipilä et al. (2014). Approximately linear pressure dependence observed by Drozd and Donahue (2011), with $Y \approx 0.05$ at 110 Torr; [k] Based on Winterhalter et al. (2000) with support from theoretical study of Nguyen et al. (2009a); [l] Based on Sipilä et al. (2014); [m] Based on Winterhalter et al. (2009) with support from theoretical study of Nguyen et al. (2009b); [n] Based on Hatekayama et al. (1984), Drozd and Donahue (2011), who observed no stabilization at 550-640 Torr, and Campos-Pineda and Zhang (2018) who observed no stabilization at 10-20 Torr; [o] Based on Drozd and Donahue (2011). Full stabilisation observed at pressures above ~ 400 Torr, with $Y$ falling at lower pressures to ~ 0.6 at 70 Torr; [p] Based on Hakala and Donahue (2018). Pressure dependence measurements suggest $Y$ falls to ~ 0.35 at 50 Torr.





**Table 3. Summary of recommended HO yields for reactions of O₃ with alkenes at 298 K and 1 bar**

| Reaction ID [a] | alkene | HO yield | comments |
|---|---|---|---|
| *Small alkene reactions* | | | |
| Ox_VOC5 | ethene | $0.17 \pm 0.05$ | (b) |
| Ox_VOC6 | propene | $0.36 \pm 0.04$ | (c) |
| Ox_VOC16 | but-1-ene | $0.38 \pm 0.18$ | (d) |
| Ox_VOC17 | *cis*-but-2-ene | $0.33 \pm 0.07$ | (e) |
| Ox_VOC18 | *trans*-but-2-ene | $0.60 \pm 0.06$ | (f) |
| Ox_VOC15 | 2-methylpropene | $0.69 \pm 0.15$ | (g) |
| Ox_VOC41 | 2,3-dimethylbut-2-ene | $0.93 \pm 0.14$ | (h) |
| Ox_VOC7 | isoprene | $0.26 \pm 0.04$ | (i) |
| *Monoterpene reactions* | | | |
| Ox_VOC8 | α-pinene | $0.80 \pm 0.10$ | (j) |
| Ox_VOC19 | β-pinene | $0.30 \pm 0.06$ | (k) |
| Ox_VOC20 | limonene | $0.66 \pm 0.04$ | (l) |
| Ox_VOC21 | camphene | $\leq 0.18$ | (m) |
| Ox_VOC22 | 2-carene | $0.81 \pm 0.11$ | (n) |
| Ox_VOC23 | 3-carene | $0.86 \pm 0.11$ | (n) |
| Ox_VOC24 | β-myrcene | $0.63 \pm 0.09$ | (n) |
| Ox_VOC25 | β-ocimene | $0.55 \pm 0.09$ | (n) |
| Ox_VOC26 | α-phellandrene | $0.29 \pm 0.05$ | (o) |
| Ox_VOC27 | β-phellandrene | $0.14\,^{+0.07}_{-0.05}$ | (m) |
| Ox_VOC28 | sabinene | $0.33 \pm 0.05$ | (n) |
| Ox_VOC29 | α-terpinene | $0.32 \pm 0.06$ | (p) |
| Ox_VOC30 | γ-terpinene | $0.81 \pm 0.11$ | (n) |
| Ox_VOC31 | terpinolene | $0.70 \pm 0.08$ | (q) |
| *Sesquiterpene reactions* | | | |
| Ox_VOC32 | β-caryophyllene | $0.08 \pm 0.03$ | (r) |
| Ox_VOC33 | α-cedrene | $0.65 \pm 0.05$ | (s) |
| Ox_VOC34 | α-copaene | $0.35\,^{+0.18}_{-0.12}$ | (t) |
| Ox_VOC37 | α-humulene | $0.16 \pm 0.06$ | (u) |

**Comments**

[a] See corresponding data sheets in Supplement A for further information; [b] Based on Atkinson et al. (1992), Paulson et al. (1999), Rickard et al. (1999), Mihelcic et al. (1999), Fenske et al. (2000) and Alam et al. (2011). Comparable pressure-independent yield (0.14) reported by Kroll et al. (2001a) over pressure range 13-80 mbar; [c] Based on Atkinson and Aschmann (1993), Neeb and Moortgat (1999), Paulson et al. (1999), Rickard et al. (1999), Aschmann et al. (2003), Qi et al. (2009) and Alam et al. (2013); [d] Based on Atkinson and Aschmann (1993), Paulson et al. (1999), Fenske et al. (2000) and Alam et al. (2013); [e] Based on Atkinson and Aschmann (1993), McGill et al. (1999), Orzechowska and Paulson (2002) and Alam et al. (2013); [f] Based on Atkinson and Aschmann (1993), McGill et al. (1999), Orzechowska and Paulson (2002), Hasson et al. (2003) and Alam et al. (2013); [g] Based on Atkinson and Aschmann (1993), Neeb and Moortgat (1999), Paulson et al. (1999), Rickard et al. (1999) and Alam et al. (2013); [h] Based on Chew and Atkinson (1996), Rickard et al. (1999), Fenske et al. (2000), Siese et al. (2001), Orzechowska and Paulson (2002), Aschmann et al. (2003) and Alam et al. (2013); [i] Based on Aschmann et al. (1996), Paulson et al. (1998), Neeb and Moortgat (1999), Malkin et al. (2010), Nguyen et al. (2016) and Ren et al. (2017); [j] Based on Atkinson et al. (1992), Chew and Atkinson (1996), Paulson et al. (1998), Rickard et al. (1999), Siese et al. (2001), Aschmann et al. (2002), Berndt et al. (2003), Presto and Donahue (2004) and Forester and Wells (2011); [k] Based on Atkinson et al. (1992) and Rickard et al. (1999); [l] Based on Aschmann et al. (2002), Herrmann et al. (2010) and Forester and Wells (2011); [m] Based on Atkinson et al. (1992); [n] Based on Aschmann et al. (2002); [o] Based on Herrmann et al. (2010); [p] Based on Aschmann et al. (2002) and Herrmann et al. (2010); [q] Based on Aschmann et al. (2002) and Herrmann et al. (2010); [r] Based on Shu and Atkinson (1994), Winterhalter et al. (2009) and Jenkin et al. (2012); [s] Based on Shu and Atkinson (1994) and Yao et al. (2014). Substantially lower yield, $0.090 \pm 0.016$, reported in the presence sCI scavengers, CH₃C(O)OH or SO₂, by Yao et al. (2014); [t] Based on Shu and Atkinson (1994); [u] Based on Shu and Atkinson (1994) and Beck et al. (2011).



**Table 4. Summary of the recommended spectral data for $C_1$ - $C_3$ sCIs**

| Reaction ID [a] | reaction | $\sigma_{max}$ $cm^2$ molecule$^{-1}$ | $\Delta\sigma_{max}$ $cm^2$ molecule$^{-1}$ | $\lambda_{max}$ nm | $\lambda$ range nm | $\phi$[b] |
|---|---|---|---|---|---|---|
| P33 | $CH_2OO + h\nu$ | $1.23 \times 10^{-17}$ | $\pm (0.18 \times 10^{-17})$ | 340 | 280-455 | 1.0 |
| P34 | $Z$-$CH_3CHOO + h\nu$ | $1.20 \times 10^{-17}$ | $\pm (0.18 \times 10^{-17})$ | 323 | 300-430 | 1.0 |
|  | $E$-$CH_3CHOO + h\nu$ | $1.20 \times 10^{-17}$ | $\pm (0.18 \times 10^{-17})$ | 360 | 300-430 | 1.0 |
| P35 | $(CH_3)_2COO + h\nu$ | $1.75 \times 10^{-17}$ | $\pm (0.53 \times 10^{-17})$ | 330 | 280-405 | 1.0 |
| P36[c] | $CH_3CH_2CHOO + h\upsilon$ | no recommendation (see data sheet) | | 322 | 280-400 | 1.0 |

**Comments**

[a] See corresponding data sheets in Supplement B Sect. B5 for further information.

[b] $\phi$ is the photodissociation quantum yield.

[c] Data sheet for $CH_3CH_2CHOO$ included for completeness, although thermal reactions of this sCI are not included in the current evaluation.





**Table 5. Summary of recommended rate coefficients for gas phase bimolecular reactions of sCIs**

| Reaction ID | Reaction | $k_{298}$ cm$^3$ molecule$^{-1}$ s$^{-1}$ | $\Delta\log k_{298}$ | $k(T)$ cm$^3$ molecule$^{-1}$ s$^{-1}$ | T range | $\Delta(E/R)$ K |
|---|---|---|---|---|---|---|
| \multicolumn{7}{l}{*Reactions of CH$_2$OO – based on data sheets in Supplement B Sect. B1*} | | | | | | |
| CGI_1 | CH$_2$OO + SO$_2$ | $3.7 \times 10^{-11}$ | $\pm 0.05$ | | | |
| CGI_2 | CH$_2$OO + NO$_2$ | $3 \times 10^{-12}$ | $\pm 0.5$ | | | |
| CGI_3 | CH$_2$OO + NO | $< 6 \times 10^{-14}$ | | | | |
| CGI_4 | CH$_2$OO + H$_2$O | $2.8 \times 10^{-16}$ | $\pm 0.3$ | | | |
| | CH$_2$OO + (H$_2$O)$_2$ | $6.4 \times 10^{-12}$ | $\pm 0.2$ | $7.35 \times 10^{-18} \exp(4076/T)$ | 280-325 | $\pm 500$ |
| CGI_5 | CH$_2$OO + CH$_2$OO | $7.4 \times 10^{-11}$ | $\pm 0.1$ | | | |
| CGI_6 | CH$_2$OO + I | $9.0 \times 10^{-12}$ | $\pm 0.3$ | | | |
| CGI_7 | CH$_2$OO + CH$_3$C(O)H | $k_0 = 1.6 \times 10^{-29}$ [M] | $\pm 0.2$ | | | |
| | | $k_\infty = 1.7 \times 10^{-12}$ | $\pm 0.2$ | | | |
| CGI_8 | CH$_2$OO + CH$_3$C(O)CH$_3$ | $3.4 \times 10^{-13}$ | $\pm 0.3$ | | | |
| CGI_9 | CH$_2$OO + CF$_3$C(O)CF$_3$ | $3.2 \times 10^{-11}$ | $\pm 0.1$ | | | |
| CGI_11 | CH$_2$OO + HC(O)OH | $1.1 \times 10^{-10}$ | $\pm 0.1$ | | | |
| CGI_10 | CH$_2$OO + CH$_3$C(O)OH | $1.3 \times 10^{-10}$ | $\pm 0.1$ | | | |
| CGI_23 | CH$_2$OO + CF$_3$C(O)OH | $3.3 \times 10^{-10}$ | $\pm 0.2$ | $3.8 \times 10^{-18} T^2 \exp(1620/T) + 2.5 \times 10^{-10}$ | 240-340 | $\pm 500$ |
| \multicolumn{7}{l}{*Reactions of Z- and E-CH$_3$CHOO – based on data sheets in Supplement B Sect. B2*} | | | | | | |
| CGI_15 | Z-CH$_3$CHOO + SO$_2$ | $2.6 \times 10^{-11}$ | $\pm 0.1$ | | | |
| | E-CH$_3$CHOO + SO$_2$ | $1.4 \times 10^{-10}$ | $\pm 0.3$ | | | |
| CGI_16 | Z-CH$_3$CHOO + H$_2$O | $< 2 \times 10^{-16}$ | | | | |
| | E-CH$_3$CHOO + H$_2$O | $1.3 \times 10^{-14}$ | $\pm 0.3$ | | | |
| | Z-CH$_3$CHOO + (H$_2$O)$_2$ | - | | | | |
| | E-CH$_3$CHOO + (H$_2$O)$_2$ | $4.4 \times 10^{-11}$ | $\pm 0.5$ | | | |
| CGI_17 | Z-CH$_3$CHOO + NO$_2$ | $2.0 \times 10^{-12}$ | $\pm 0.15$ | | | |
| | E-CH$_3$CHOO + NO$_2$ | $2.0 \times 10^{-12}$ | $\pm 0.3$ | | | |
| CGI_26 | Z-CH$_3$CHOO + HC(O)OH | $2.5 \times 10^{-10}$ | $\pm 0.1$ | | | |
| | E-CH$_3$CHOO + HC(O)OH | $5.0 \times 10^{-10}$ | $\pm 0.3$ | | | |
| CGI_27 | Z-CH$_3$CHOO + CH$_3$C(O)OH | $1.7 \times 10^{-10}$ | $\pm 0.15$ | | | |
| | E-CH$_3$CHOO + CH$_3$C(O)OH | $2.5 \times 10^{-10}$ | $\pm 0.15$ | | | |
| \multicolumn{7}{l}{*Reactions of (CH$_3$)$_2$COO – based on data sheets in Supplement B Sect. B3*} | | | | | | |
| CGI_18 | (CH$_3$)$_2$COO + SO$_2$ | $k_\infty = 1.55 \times 10^{-10}$ | $\pm 0.15$ | $k_\infty = 4.23 \times 10^{-13} \exp(1760/T)$ | 280-305 | $\pm 500$ |
| CGI_19 | (CH$_3$)$_2$COO + H$_2$O | $< 1.5 \times 10^{-16}$ | | | | |
| | (CH$_3$)$_2$COO + (H$_2$O)$_2$ | $< 1.3 \times 10^{-13}$ | | | | |
| CGI_20 | (CH$_3$)$_2$COO + NO$_2$ | $2.1 \times 10^{-12}$ | $\pm 0.3$ | | | |
| CGI_28 | (CH$_3$)$_2$COO + HC(O)OH | $3.1 \times 10^{-10}$ | $\pm 0.1$ | | | |
| CGI_29 | (CH$_3$)$_2$COO + CH$_3$C(O)OH | $3.1 \times 10^{-10}$ | $\pm 0.1$ | | | |
| CGI_24 | (CH$_3$)$_2$COO + CF$_3$C(O)OH | $6.2 \times 10^{-10}$ | $\pm 0.2$ | $4.9 \times 10^{-18} T^2 \exp(1620/T) + 5.2 \times 10^{-10}$ | 240-340 | $\pm 500$ |





**Table 5 (continued). Summary of recommended rate coefficients for gas phase bimolecular reactions of sCIs**

| Reaction ID | Reaction | $k_{298}$ cm$^3$ molecule$^{-1}$ s$^{-1}$ | $\Delta\log k_{298}$ | $k(T)$ cm$^3$ molecule$^{-1}$ s$^{-1}$ | $T$ range | $\Delta(E/R)$ K |
|---|---|---|---|---|---|---|
| *Reactions of C$_4$intermediates from isoprene – based on data sheets in Supplement B Sect. B4* | | | | | | |
| CGI_21 [a] | *Z*-(CH=CH$_2$)(CH$_3$)COO + SO$_2$ | $1.55 \times 10^{-10}$ | - | $4.23 \times 10^{-13}$ exp(1760/*T*) | - | - |
| | *Z*-(CH=CH$_2$)(CH$_3$)COO + H$_2$O | $1.79 \times 10^{-18}$ | - | $2.21 \times 10^{-21}$ $T^{2.27}$ exp(-1858/*T*) | - | - |
| | *Z*-(CH=CH$_2$)(CH$_3$)COO + (H$_2$O)$_2$ | $4.87 \times 10^{-15}$ | - | $2.25 \times 10^{-21}$ $T^{2.27}$ exp(493/*T*) | - | - |
| | *E*-(CH=CH$_2$)(CH$_3$)COO + SO$_2$ | $1.55 \times 10^{-10}$ | - | $4.23 \times 10^{-13}$ exp(1760/*T*) | - | - |
| | *E*-(CH=CH$_2$)(CH$_3$)COO + H$_2$O | $7.89 \times 10^{-20}$ | - | $7.07 \times 10^{-19}$ $T^{1.46}$ exp(-3132/*T*) | - | - |
| | *E*-(CH=CH$_2$)(CH$_3$)COO + (H$_2$O)$_2$ | $3.06 \times 10^{-16}$ | - | $7.63 \times 10^{-19}$ $T^{1.45}$ exp(-675/*T*) | - | - |
| | *Z*-(C(CH$_3$)=CH$_2$)CHOO + SO$_2$ | $2.6 \times 10^{-11}$ | - | | - | - |
| | *Z*-(C(CH$_3$)=CH$_2$)CHOO + H$_2$O | $1.19 \times 10^{-19}$ | - | $2.24 \times 10^{-19}$ $T^{1.65}$ exp(-2989/*T*) | - | - |
| | *Z*-(C(CH$_3$)=CH$_2$)CHOO + (H$_2$O)$_2$ | $4.39 \times 10^{-16}$ | - | $2.42 \times 10^{-19}$ $T^{1.64}$ exp(-548/*T*) | - | - |
| | *E*-(C(CH$_3$)=CH$_2$)CHOO + SO$_2$ | $1.4 \times 10^{-10}$ | - | | - | - |
| | *E*-(C(CH$_3$)=CH$_2$)CHOO + H$_2$O | $1.43 \times 10^{-16}$ | - | $2.93 \times 10^{-19}$ $T^{1.66}$ exp(-973/*T*) | - | - |
| | *E*-(C(CH$_3$)=CH$_2$)CHOO + (H$_2$O)$_2$ | $2.79 \times 10^{-13}$ | - | $3.24 \times 10^{-19}$ $T^{1.65}$ exp(1271/*T*) | - | - |
| CGI_25 | *E*-(CH=CH$_2$)(CH$_3$)COO + CF$_3$C(O)OH | $7.3 \times 10^{-10}$ | ± 0.3 | $4.9 \times 10^{-18}$ $T^2$ exp(1620/*T*) + $6.3 \times 10^{-10}$ | 240 -340 | ± 500 |
| | *E*-(C(CH$_3$)=CH$_2$)CHOO + CF$_3$C(O)OH | $7.3 \times 10^{-10}$ | ± 0.3 | $4.9 \times 10^{-18}$ $T^2$ exp(1620/*T*) + $6.3 \times 10^{-10}$ | 240 -340 | ± 500 |

**Comments:** [a] Rate coefficients for SO$_2$ reactions are inferred from recommendations for (CH$_3$)$_2$COO for *Z*- and *E*-(CH=CH$_2$)(CH$_3$)COO, *Z*-CH$_3$CHOO for *Z*-(C(CH$_3$)=CH$_2$)CHOO, and *E*-CH$_3$CHOO for *E*-(C(CH$_3$)=CH$_2$)CHOO. Temperature-dependent rate coefficients for H$_2$O and (H$_2$O)$_2$ reactions are adopted from the theoretical/SAR methods reported by Vereecken et al. (2017), as presented in Supplement Tables 35 and 40 of that paper. Individual parameters are not currently assigned uncertainties, but performance of ensemble of reactions (also including sCI decomposition reactions) was tested against reported O$_3$ + isoprene product observations (see data sheet CGI_21, Supplement B).



**Table 6. Summary of recommended rate coefficients for gas phase unimolecular reactions of sCIs**

| Reaction ID | Reaction | $k_{298}$ s$^{-1}$ | $\Delta \log k_{298}$ | $k(T)$ s$^{-1}$ | $T$ range | $\Delta(E/R)$ K |
|---|---|---|---|---|---|---|
| *Based on data sheets in Supplement B Sects. B1-B4* | | | | | | |
| CGI_12 | $CH_2OO + M$ | $1 \times 10^{-3}$ (1 bar) | ± 1.0 | $k_0 = 3.2 \times 10^{-4} (T/298)^{-5.81} \exp(-12770/T)[M]$ | 450-650 | ± 500 |
| | | | | $k_\infty = 1.4 \times 10^{13}(T/298)^{0.06} \exp(-10010/T)$ | 450-650 | ± 500 |
| | | | | $(F_c = 0.447)$ | | |
| CGI_13 | $Z$-$CH_3CHOO + M$ | 150 (1 bar) | ± 0.3 | $7.4 \times 10^6 \exp(-3220/T)$ | 275-320 | ± 700 |
| | $E$-$CH_3CHOO + M$ | 60 (1 bar) | ± 0.5 | | | |
| CGI_14 | $(CH_3)_2COO + M$ | 400 (1 bar) | ± 0.2 | $7.2 \times 10^6 \exp(-2920/T)$ | 280-330 | ± 700 |
| CGI_21 [a] | $Z$-$(CH{=}CH_2)(CH_3)COO + M$ | 13,600 | - | $9.75 \times 10^8 T^{1.03} \exp(-5081/T)$ | - | - |
| | $E$-$(CH{=}CH_2)(CH_3)COO + M$ | 51.3 | - | $4.36 \times 10^{-67} T^{25.9} \exp(2737/T)$ | - | - |
| | $Z$-$(C(CH_3){=}CH_2)CHOO + M$ | 14,000 | - | $2.58 \times 10^9 T^{0.87} \exp(-5090/T)$ | - | - |
| | $E$-$(C(CH_3){=}CH_2)CHOO + M$ | 30.2 | - | $1.68 \times 10^{10} T^{1.02} \exp(-7732/T)$ | - | - |

**Comments:** [a] Temperature-dependent rate coefficients adopted from the theoretical/SAR methods reported by Vereecken et al. (2017), as presented in Supplement Table 31 of that paper (N.B. exponent of the pre-exponential factor changed from 9 to 8 in the case of $Z$-$(CH{=}CH_2)(CH_3)COO$, for consistency with 298 K rate coefficient reported by Vereecken et al. (2017)). Individual parameters are not currently assigned uncertainties, but performance of the ensemble of reactions (also including sCI reactions with $SO_2$, $H_2O$ and $(H_2O)_2$) was tested against reported $O_3$ + isoprene product observations (see data sheet CGI_21, Supplement B).





**Table 7. Rate coefficients ($k$) and representative lower tropospheric first order loss rates ($k_I$) at 298 K for sCI bimolecular reactions with $SO_2$, $H_2O$ and $(H_2O)_2$, and unimolecular decomposition. The present recommendations (IUPAC) are compared with calculated values using the theory-based SAR developed by Vereecken et al. (2017).**

| Reaction ID | Reaction | $k$ [a] | | $k_I$ (s$^{-1}$) [b] | |
|---|---|---|---|---|---|
| | | IUPAC | SAR [c] | IUPAC | SAR [c] |
| *Reactions of CH$_2$OO* | | | | | |
| CGI_1 | $CH_2OO + SO_2$ | $3.7 \times 10^{-11}$ | - | 0.93 | - |
| CGI_4 | $CH_2OO + H_2O$ | $2.8 \times 10^{-16}$ | $8.63 \times 10^{-16}$ | 86 | 266 |
| | $CH_2OO + (H_2O)_2$ | $6.4 \times 10^{-12}$ | $1.48 \times 10^{-12}$ | 1250 | 289 |
| CGI_12 | $CH_2OO + M$ | $1 \times 10^{-3}$ s$^{-1}$ | $2.80 \times 10^{-1}$ s$^{-1}$ | 0.001 | 0.280 |
| *Reactions of Z-CH$_3$CHOO* | | | | | |
| CGI_15 | $Z$-$CH_3CHOO + SO_2$ | $2.6 \times 10^{-11}$ | - | 0.65 | - |
| CGI_16 | $Z$-$CH_3CHOO + H_2O$ | $< 2 \times 10^{-16}$ | $6.84 \times 10^{-19}$ | $< 62$ | 0.210 |
| | $Z$-$CH_3CHOO + (H_2O)_2$ | - | $2.05 \times 10^{-15}$ | - | 0.401 |
| CGI_13 | $Z$-$CH_3CHOO + M$ | $1.5 \times 10^2$ s$^{-1}$ | $1.37 \times 10^2$ s$^{-1}$ | 150 | 137 |
| *Reactions of E-CH$_3$CHOO* | | | | | |
| CGI_15 | $E$-$CH_3CHOO + SO_2$ | $1.4 \times 10^{-10}$ | - | 3.5 | - |
| CGI_16 | $E$-$CH_3CHOO + H_2O$ | $1.3 \times 10^{-14}$ | $2.33 \times 10^{-14}$ | 4000 | 7190 |
| | $E$-$CH_3CHOO + (H_2O)_2$ | $4.4 \times 10^{-11}$ | $2.63 \times 10^{-11}$ | 8600 | 5150 |
| CGI_13 | $E$-$CH_3CHOO + M$ | $6.0 \times 10^1$ s$^{-1}$ | $5.22 \times 10^1$ s$^{-1}$ | 60 | 52.2 |
| *Reactions of (CH$_3$)$_2$COO* | | | | | |
| CGI_18 | $(CH_3)_2COO + SO_2$ | $1.55 \times 10^{-10}$ | - | 3.9 | - |
| CGI_19 | $(CH_3)_2COO + H_2O$ | $< 1.5 \times 10^{-16}$ | $7.40 \times 10^{-18}$ | $< 46$ | 2.28 |
| CGI_19 | $(CH_3)_2COO + (H_2O)_2$ | $< 1.3 \times 10^{-13}$ | $1.79 \times 10^{-14}$ | $< 25$ | 3.51 |
| CGI_14 | $(CH_3)_2COO + M$ | $4.0 \times 10^2$ s$^{-1}$ | $5.01 \times 10^2$ s$^{-1}$ | 400 | 501 |
| *Reactions of C$_4$ intermediates from isoprene* | | | | | |
| CGI_21 | $Z$-$(CH{=}CH_2)(CH_3)OO + SO_2$ | $1.55 \times 10^{-10}$ [d] | - | 3.9 | - |
| | $Z$-$(CH{=}CH_2)(CH_3)OO + H_2O$ | [e] | $1.79 \times 10^{-18}$ | [e] | 0.551 |
| | $Z$-$(CH{=}CH_2)(CH_3)OO + (H_2O)_2$ | [e] | $4.87 \times 10^{-15}$ | [e] | 0.951 |
| | $Z$-$(CH{=}CH_2)(CH_3)OO + M$ | [e] | $1.36 \times 10^4$ s$^{-1}$ | [e] | 13600 |
| | $E$-$(CH{=}CH_2)(CH_3)OO + SO_2$ | $1.55 \times 10^{-10}$ [d] | - | 3.9 | - |
| | $E$-$(CH{=}CH_2)(CH_3)OO + H_2O$ | [e] | $7.89 \times 10^{-20}$ | [e] | 0.0243 |
| | $E$-$(CH{=}CH_2)(CH_3)OO + (H_2O)_2$ | [e] | $3.06 \times 10^{-16}$ | [e] | 0.0599 |
| | $E$-$(CH{=}CH_2)(CH_3)OO + M$ | [e] | $5.13 \times 10^1$ s$^{-1}$ | [e] | 51.3 |
| | $Z$-$(C(CH_3){=}CH_2)CHOO + SO_2$ | $2.6 \times 10^{-11}$ [d] | - | 0.65 | - |
| | $Z$-$(C(CH_3){=}CH_2)CHOO + (H_2O)_2$ | [e] | $1.19 \times 10^{-19}$ | [e] | 0.0367 |
| | $Z$-$(C(CH_3){=}CH_2)CHOO + SO_2$ | [e] | $4.39 \times 10^{-16}$ | [e] | 0.0859 |
| | $Z$-$(C(CH_3){=}CH_2)CHOO + M$ | [e] | $1.40 \times 10^4$ s$^{-1}$ | [e] | 14000 |
| | $E$-$(C(CH_3){=}CH_2)CHOO + SO_2$ | $1.4 \times 10^{-10}$ [d] | - | 3.5 | - |
| | $E$-$(C(CH_3){=}CH_2)CHOO + (H_2O)_2$ | [e] | $1.43 \times 10^{-16}$ | [e] | 44.1 |
| | $E$-$(C(CH_3){=}CH_2)CHOO + SO_2$ | [e] | $2.79 \times 10^{-13}$ | [e] | 54.5 |
| | $E$-$(C(CH_3){=}CH_2)CHOO + M$ | [e] | $3.02 \times 10^1$ s$^{-1}$ | [e] | 30.2 |

**Comments:** [a] Units of $k$ are cm$^3$ molecule$^{-1}$ s$^{-1}$, unless otherwise stated; [b] $k_I$ at 298 K and 1 bar determined for 40 % relative humidity ($[H_2O]$ = $3.08 \times 10^{17}$ molecule cm$^{-3}$ and $[(H_2O)_2]$ = $1.96 \times 10^{14}$ molecule cm$^{-3}$), and for $[SO_2]$ = $2.5 \times 10^{10}$ molecule cm$^{-3}$ (~1 ppbv); [c] Based on $T$-dependence parameters given in Tables 31, 35 and 40 of Vereecken et al. (2017); [d] Value inferred from those for C$_2$ and C$_3$ sCIs (see Table 5, comment (a)); [e] IUPAC $k$ value is adopted from Vereecken et al. (2017), and given entry is therefore identical to the SAR value shown.





**Table 8.** Summary of results for the representative ambient calculations (see Sect. 8 and Supplement C for further details).

| | Rural background [a] | | Suburban background [b] | | Urban traffic [c] | |
|---|---|---|---|---|---|---|
| | winter | summer | winter | summer | winter | summer |
| sCI production rate (molecule $cm^{-3}$ $s^{-1}$) | $6.25 \times 10^4$ | $8.77 \times 10^4$ | $8.63 \times 10^4$ | $2.53 \times 10^5$ | $6.53 \times 10^4$ | $2.17 \times 10^5$ |
| sCI concentration (molecule $cm^{-3}$) | 389 | 381 | 539 | 1102 | 384 | 886 |
| $SO_2$ oxidation rate (% $h^{-1}$) | 0.006 | 0.008 | 0.009 | 0.031 | 0.006 | 0.009 |
| HC(O)OH oxidation rate (% $h^{-1}$) | 0.04 | 0.04 | 0.05 | 0.11 | 0.04 | 0.08 |
| ***Contributions to total sCI loss*** | | | | | | |
| Unimolecular decomposition | 46.9% | 50.6% | 48.0% | 59.1% | 45.5% | 49.9% |
| Reaction with $H_2O$ | 9.4% | 9.7% | 9.1% | 8.7% | 8.7% | 9.4% |
| Reaction with $(H_2O)_2$ | 43.0% | 36.8% | 40.6% | 27.6% | 40.9% | 36.0% |
| Reaction with $SO_2$ | 0.3% | 0.2% | 1.4% | 1.6% | 1.3% | 0.6% |
| Reaction with $NO_2$ | 0.2% | 0.1% | 0.4% | 0.1% | 1.1% | 0.6% |
| Reaction with HC(O)OH | 0.2% | 2.7% | 0.5% | 2.9% | 2.5% | 3.5% |
| ***Contributions to sCI production rate*** | | | | | | |
| Core $C_1$-$C_4$ set [c] | 90.9% | 74.4% | 88.4% | 42.3% | 87.7% | 84.9% |
| Others $C_3$-$C_5$ (anthropogenic) [d] | 9.1% | 7.2% | 11.6% | 4.6% | 12.3% | 7.5% |
| Others $C_{10}$ (biogenic) [e] | - | 18.4% | - | 53.1% | - | 7.7% |
| ***Contributions to sCI concentration*** | | | | | | |
| Core $C_1$-$C_4$ set [c] | 89.3% | 67.6% | 86.6% | 34.2% | 85.8% | 83.1% |
| Others $C_3$-$C_5$ (anthropogenic) [d] | 10.7% | 7.5% | 13.4% | 4.5% | 14.2% | 7.7% |
| Others $C_{10}$ (biogenic) [e] | - | 24.9% | - | 61.3% | - | 9.2% |
| ***Contributions to $SO_2$ oxidation rate*** [f] | | | | | | |
| Core $C_1$-$C_4$ set [c] | 91.7% | 41.4% | 91.1% | 18.4% | 91.0% | 63.8% |
| Others $C_3$-$C_5$ (anthropogenic) [d] | 8.3% | 3.5% | 8.9% | 1.1% | 9.0% | 4.6% |
| Others $C_{10}$ (biogenic) [e] | - | 55.2% | - | 80.6% | - | 31.7% |
| ***Contributions to HC(O)OH oxidation rate*** [f] | | | | | | |
| Core $C_1$-$C_4$ set [c] | 88.8% | 64.5% | 86.6% | 31.6% | 85.9% | 81.7% |
| Others $C_3$-$C_5$ (anthropogenic) [d] | 11.2% | 7.5% | 13.4% | 4.0% | 14.1% | 7.6% |
| Others $C_{10}$ (biogenic) [e] | - | 28.0% | - | 64.5% | - | 10.7% |

**Comments:** [a] Based on data for the Chilbolton Observatory site (51.149617, -1.438228); [b] Based on data for the London Eltham site (51.452580, 0.070766); [b] Based on data for the London Marylebone Road site (51.522530, -0.154611); [c] The core $C_1$-$C_4$ set comprises the sCIs shown in Fig. 2, for which evaluated rate parameters have been presented in Sect. 6; [d] Comprises *E*- and *Z*-$C_2H_5$CHOO, *E*- and *Z*-*n*-$C_3H_7$CHOO, *E*- and *Z*-*i*-$C_3H_7$CHOO, *E*- and *Z*-*n*-$C_4H_9$CHOO, *E*- and *Z*-($C_2H_5$)($CH_3$)COO, *E*- and *Z*-($CH=CH_2$)CHOO formed from the $C_4$-$C_6$ alkenes considered (see Supplement C); [e] Comprises eight pinonaldehyde oxide and limononaldehyde oxide isomers formed from the monoterpenes α-pinene and limonene (see Supplement C); [f] Specifies the contribution to the total $SO_2$ or HC(O)OH oxidation rate due to reaction with sCIs.