# Peer review of "Evaluated kinetic and photochemical data for atmospheric chemistry: Volume VII - Criegee intermediates"

_Atmospheric Chemistry and Physics, 2020_

## Referee Comment (RC1) · Craig Taatjes (Referee) · 7 Jul 2020

This work assesses kinetics for the fast-moving field of carbonyl oxide chemistry. The evaluation is thorough and thoughtful and accurately captures and accounts for most of the questions and disagreements that arise from the available studies. The evaluation makes insightful recommendations of the key points for additional research (including branching fractions for some key reactions). The manuscript, although an experiment-based evaluation, also points out the key role of advanced theoretical methods, which now can produce highly accurate kinetics values for experimentally inaccessible reactions. This is an excellent work that will be very useful to modelers and as a guide

to experimental kineticists. There are just a few improvements that I would suggest, focused on the aspect I am most familiar with, reaction kinetics of carbonyl oxides.

First, new investigations have continued since the evaluation that forms the basis of this paper, especially in the reaction kinetics of the stabilized carbonyl oxides. That is unavoidable. However, the authors may consider the newest investigations (PNAS 117 (18), 9733-9740 (2020)) of the kinetics of the C4 carbonyl oxide methyl vinyl ketone oxide, because here theory and direct experiments suggest that the bimolecular reactions of the conjugated carbonyl oxides may differ from the reactions used as analogies in the evaluation. For example, direct kinetics measurements show that the reaction of methyl vinyl ketone oxide with SO2 is a factor of three or so slower than the reactions of non-conjugated carbonyl oxides of similar size.

Second, as the manuscript acknowledges, measuring thermal kinetics for unimolecular reactions of carbonyl oxides is often experimentally difficult because of the need to correct for competing bimolecular processes. In this case the theoretical characterizations, especially those for which microcanonical rate coefficients and the related tunneling parameters have been corroborated by direct energy-specific dynamics measurements, provide valuable information about the unimolecular processes (see Int. Rev. Phys. Chem. 39 (1), 1-33 (2020)). The evaluation already makes good use of theory in interpreting the unimolecular kinetics, but the authors might consider emphasizing experimentally validated calculations (J. Chem. Phys. 146, 134307 (2017)) for acetone oxide (k298 = 276 s-1), where the tunneling-adjusted microcanonical rate coefficients match experiment over a wide energy range, and J. Chem. Phys. 145, 234308 (2016) that treats the unimolecular decay of Z-acetaldehyde oxide with similar tests of tunneling parameters against direct experiment (k298 = 122 s-1). These values are slightly lower than the recommendations. I note that the unimolecular decay kinetics for some methyl vinyl ketone oxide conformers (J. Am. Chem. Soc., 140 (34), 10866–10880 (2018)) have also been compared to energy-resolved dynamics measurements.

Third, in the discussion of the atmospheric role of carbonyl oxides, are there other nonkinetics uncertainties (e.g., in the alkene source inventory) that should be mentioned? The development of sensitive measurement methods for carbonyl oxides (e.g., J. Am. Chem. Soc. 139 (38), 13387–13392 (2017)) that may eventually constrain the concentration of these intermediates in the field should possibly be mentioned as an important area for continued effort.

Finally – is there a reference to verify a unity quantum yield of O (1D) from UV excitation of carbonyl oxides (section 5)?
* * *

---

## Referee Comment (RC2) · Anonymous Referee #2 · 14 Sep 2020

The manuscript provides a comprehensive assessment of gas-phase Criegee intermediate chemistry and photochemistry. Kinetics measurements are summarized thoroughly for ozonolysis of a broad range of unsaturated VOCs, the reactions of stabilized Criegee intermediates with a selection of trace gases, and unimolecular decomposition reactions. Where appropriate, theoretical work is also referenced to support the assessment of experimental studies. Overall, this is an excellent and thorough summary of our current understanding of Criegee intermediate chemistry in the atmosphere.

I have only very minor suggestions for the authors. First, the discovery of the UV spectrum of formaldehyde oxide that is attributed to Sheps [J. Phys. Chem. Lett. 4, 4201

(2013)] on page 10 should more properly be attributed to Beames et al. [J. Am. Chem. Soc. 134, 20045 (2012)]. Second, while discussing on page 9 apparent discrepancies between measurements of the UV absorption spectrum the authors comment on the lack of detailed data on the temperature dependence of the cross sections. Foreman et al. [Phys. Chem. Chem. Phys. 17, 32539 (2015)] demonstrated that the spectra were independent of temperature over the range 276-357 K. Third, the range of reactions of stabilized Criegee intermediates covered in the assessment is somewhat smaller than that compiled by Khan et al. [Environ. Sci.: Processes Impacts 20, 437 (2018)]. The authors may want to comment explicitly on why they have focused on the more limited set of reactions.

---

## Author Response (AR1)

**Authors' responses to referee and discussion comments on:** Cox et al., Atmos. Chem. Phys. Discuss., https://doi.org/10.5194/acp-2020-472.

We are very grateful to the referees for their supportive comments on this work, and for their helpful suggestions for modifications and improvements. Responses to the comments are now provided (the original comments are shown in blue font), along with the revised manuscript (showing tracked changes).

**A. Comments by Craig Taatjes**

**General comments**:

This work assesses kinetics for the fast-moving field of carbonyl oxide chemistry. The evaluation is thorough and thoughtful and accurately captures and accounts for most of the questions and disagreements that arise from the available studies. The evaluation makes insightful recommendations of the key points for additional research (including branching fractions for some key reactions). The manuscript, although an experiment based evaluation, also points out the key role of advanced theoretical methods, which now can produce highly accurate kinetics values for experimentally inaccessible reactions. This is an excellent work that will be very useful to modelers and as a guide to experimental kineticists. There are just a few improvements that I would suggest, focused on the aspect I am most familiar with, reaction kinetics of carbonyl oxides.

We thank the referee for these positive comments on our evaluation; and for the helpful suggestions for improvements and clarifications, which are dealt with in the responses below.

Specific comments:

**Comment A1**: First, new investigations have continued since the evaluation that forms the basis of this paper, especially in the reaction kinetics of the stabilized carbonyl oxides. That is unavoidable. However, the authors may consider the newest investigations (PNAS 117 (18), 9733-9740 (2020)) of the kinetics of the C4 carbonyl oxide methyl vinyl ketone oxide, because here theory and direct experiments suggest that the bimolecular reactions of the conjugated carbonyl oxides may differ from the reactions used as analogies in the evaluation. For example, direct kinetics measurements show that the reaction of methyl vinyl ketone oxide with SO2 is a factor of three or so slower than the reactions of non-conjugated carbonyl oxides of similar size.

Response: We are grateful to the referee for highlighting the study of Caravan et al. (2020) on the kinetics of methyl vinyl ketone (MVK) oxide, which was published at around the time our paper was submitted. We had anticipated that such studies were likely imminent (page 11, line 4) and are pleased to see that important new work on structurally-complex sCIs is already emerging.

We have since expanded the evaluation to include the reactions of *syn*-MVK oxide ($E$-(CH=CH$_2$)(CH$_3$)COO) with SO$_2$ and HC(O)OH, data sheets for which (CGI_22 and CGI_30, respectively) are now available at http://iupac.pole-ether.fr/, and included in Supplement B of the revised manuscript. Our preferred values for these rate coefficients are thus based on the determinations of Caravan et al. (2020), and appear in Table 5 in the revised manuscript. The work is now also referred to in Sects. 6.1 and 6.2 of the revised manuscript, where the kinetics of structurally complex sCIs are discussed. Thus, in Sect. 6.1 (*Measurements of absolute rate coefficients for reactions of sCIs*), the following text appears (new or adjusted text in red font):

> "As indicated in Sect. 5.2, methods for the production and spectroscopic characterization of more complex isoprene-derived species are emerging (e.g. Barber et al., 2018; Vansco et al., 2018; 2019), and these have provided the basis for their direct kinetics study (Caravan et al., 2020). However, direct experimental determinations of rate coefficients have not yet been reported for larger complex species (> C$_4$), derived for example from monoterpenes and sesquiterpenes."

and in Sect. 6.2 (*Evaluation of rate coefficients for bimolecular reactions*):

> "With exception of the reactions of $E$-(CH=CH$_2$)(CH$_3$)COO with SO$_2$ and HC(O)OH (Caravan et al., 2020), there are currently no direct kinetics determinations for reactions of the C$_4$ sCIs derived from isoprene."

This has also led to a minor revision to the analysis presented in data sheet CGI_21 on isoprene-derived sCIs (included in Supplement B), which assessed the parameters assigned to the ensemble of unimolecular decomposition reactions and bimolecular reactions with SO$_2$, H$_2$O and (H$_2$O)$_2$, using the results of published chamber and slow-flow experiments (Sipilä et al., 2014; Newland et al., 2015; Nguyen et al., 2016).

**Comment A2**: Second, as the manuscript acknowledges, measuring thermal kinetics for unimolecular reactions of carbonyl oxides is often experimentally difficult because of the need to correct for competing bimolecular processes. In this case the theoretical characterizations, especially those for which microcanonical rate coefficients and the related tunneling parameters have been corroborated by direct energy-specific dynamics measurements, provide valuable information about the unimolecular processes (see Int. Rev. Phys. Chem. 39 (1), 1-33 (2020)). The evaluation already makes good use of

theory in interpreting the unimolecular kinetics, but the authors might consider emphasizing experimentally validated calculations (J. Chem. Phys. 146, 134307 (2017)) for acetone oxide (k298 = 276 s-1), where the tunneling-adjusted microcanonical rate coefficients match experiment over a wide energy range, and J. Chem. Phys. 145, 234308 (2016) that treats the unimolecular decay of Z-acetaldehyde oxide with similar tests of tunneling parameters against direct experiment (k298 = 122 s-1). These values are slightly lower than the recommendations. I note that the unimolecular decay kinetics for some methyl vinyl ketone oxide conformers (J. Am. Chem. Soc., 140 (34), 10866–10880 (2018)) have also been compared to energy-resolved dynamics measurements..

Response: We thank the referee for alerting us to the recent review of Stephenson and Lester (2020) on unimolecular decay dynamics of Criegee intermediates, which we now refer to in Sect. 7 (*Overall reactivity conclusions – comparison of experiment and theory*) in the revised manuscript.

We also agree that the studies of Fang et al. (2016; 2017) on *Z*-CH$_3$CHOO and (CH$_3$)$_2$COO, and Barber et al. (2018) on *syn*-MVK oxide (*E*-(CH=CH$_2$)(CH$_3$)COO), should have been cited in the manuscript in relation to the comparison of energy-resolved dynamics measurements with theoretical estimates. The following information therefore now also appears Sect. 7 (new text in red font):

"However, the body of experimental information now available for a series of sCIs allows the results of theoretical studies to be validated, and for the methods to be refined, optimised and extended. This has included comparison of experimental and theoretical unimolecular decay rates of a number of infra-red activated sCIs for a range of excitation energies (e.g. Fang et al., 2016; 2017; Barber et al., 2018)."

In addition to the main manuscript, it should be noted that the results of theoretical studies were already discussed (in the "*comments on preferred values*") in all the unimolecular decay reaction data sheets in Supplement B, thereby providing an indication of where theoretical calculations have been experimentally validated. Reference to Barber et al. (2018), for example, thus also appears in data sheet CGI_21, relating to the unimolecular decay kinetics of *syn*-MVK oxide, *E*-(CH=CH$_2$)(CH$_3$)COO, where the following text appears:

"Barber et al. (2018) also report HO formation from the thermal decomposition of *E*-(CH=CH$_2$)(CH$_3$)COO, compatible with the expected pathway involving 1,4 H-atom transfer from the -CH$_3$ group to form a vinyl hydroperoxide intermediate, and at a rate that is consistent with theory."

It should also be noted that the results of a large number of theoretical studies are taken into account in the comprehensive theoretical/SAR study of Vereecken et al. (2017), which is mainly used for the experiment/theory comparisons that we present in the manuscript.

**Comment A3**: Third, in the discussion of the atmospheric role of carbonyl oxides, are there other non-kinetics uncertainties (e.g., in the alkene source inventory) that should be mentioned? The development of sensitive measurement methods for carbonyl oxides (e.g., J. Am. Chem. Soc. 139 (38), 13387–13392 (2017)) that may eventually constrain the concentration of these intermediates in the field should possibly be mentioned as an important area for continued effort.

Response: The referee raises two valid points, both of which have been addressed in the revised manuscript. An additional section has now been included at the end of Supplement C to discuss uncertainties in the steady state calculations, including potentially missing sources and sinks. The following text has also been added to the introductory paragraph of Sect. 8 (*Impact of Criegee intermediates in atmospheric oxidation chemistry*):

"Although the calculations presented here aim to take account of the most important production and loss routes, the estimates are inevitably subject to potential omissions and uncertainties in the sources and sinks of the sCIs (as discussed further in Sect. C4), in addition to uncertainties associated with the kinetic parameters and sCI yields, as discussed in earlier sections and in the data sheets in Supplements A and B."

Regarding measurement methods, the following text has now been included at the end of Sect. 8:

"In addition, the continued development of sensitive detection methods for sCIs (e.g. Berndt et al., 2017), that may eventually allow sCI concentrations to be measured in the field, would be valuable for evaluation and testing of the representation of sCI sources and sinks in atmospheric models."

**Comment A4**: Finally – is there a reference to verify a unity quantum yield of O (1D) from UV excitation of carbonyl oxides (section 5)?

Response: We agree that further information is required here. We have therefore replaced the relevant sentence in Sect. 5 with the text shown below, specifically referring to information for CH$_2$OO:

"Photodissociation of CH$_2$OO from this excitation is reported to proceed with a quantum yield of unity (e.g. Ting et al., 2014), producing HCHO and either O($^3$P) or O($^1$D). Production of O($^1$D) has been reported to be dominant (e.g. Li et al., 2015), and exclusive in the long wavelength tail ($\lambda \geq 364$ nm; Vansco et al., 2017)."

**B. Comments by Referee 2**

**Opening comment:**

The manuscript provides a comprehensive assessment of gas-phase Criegee intermediate chemistry and photochemistry. Kinetics measurements are summarized thoroughly for ozonolysis of a broad range of unsaturated VOCs, the reactions of stabilized Criegee intermediates with a selection of trace gases, and unimolecular decomposition reactions. Where appropriate, theoretical work is also referenced to support the assessment of experimental studies. Overall, this is an excellent and thorough summary of our current understanding of Criegee intermediate chemistry in the atmosphere.

I have only very minor suggestions for the authors.

Response: We thank the referee for these positive comments on our work, and for the suggestions for improvements, which are dealt with in the following responses.

**Comment B1**: First, the discovery of the UV spectrum of formaldehyde oxide that is attributed to Sheps [J. Phys. Chem. Lett. 4, 4201 (2013)] on page 10 should more properly be attributed to Beames et al. [J. Am. Chem. Soc. 134, 20045 (2012)].

Response: We thank the referee for pointing this out. Because the relevant text is discussing the application of UV absorption spectroscopy to the study of bimolecular reaction kinetics at ambient temperatures and pressures, we feel the study of the UV absorption spectrum (and kinetics) of Sheps (2013) at 295 K is a more appropriate reference at this point than the prior study of the UV photodissociation action spectrum of Beames et al. (2012) in a cooled supersonic expansion. However, the referee is quite correct that the "discovery" of the UV spectrum is more properly attributed to Beames et al. (2012). Noting that the sentence already specifies "absorption spectrum", we have therefore replaced the word "discovery" with "characterization".

As indicated below in our *additional changes* section, we have now also added reference to both Beames et al. (2012) and Sheps (2013) in an earlier section (*Sect. 5.2: UV spectra of stabilised Criegee intermediates*).

**Comment B2**: Second, while discussing on page 9 apparent discrepancies between measurements of the UV absorption spectrum the authors comment on the lack of detailed data on the temperature dependence of the cross sections. Foreman et al. [Phys. Chem. Chem. Phys. 17, 32539 (2015)] demonstrated that the spectra were independent of temperature over the range 276-357 K.

Response: The referee is correct that Foreman et al. (2015) report temperature-dependent measurements over the range 276-357 K in the long-wavelength tail for $CH_2OO$. However, the point we make remains valid, because this temperature range does not encompass the very low temperatures reached in molecular beams. We therefore now make this clear in the revised text at the appropriate point:

> "… the lack of detailed data on the temperature dependence of the cross sections over the required range precludes firm conclusions to be drawn."

As a result of the referee's comment, we realised that the lack of the temperature dependence reported by Foreman et al. (2015) was not previously explicitly stated in the relevant data sheet, P33 (in Supplement B), and a statement to this effect has now therefore been included.

**Comment B3:** Third, the range of reactions of stabilized Criegee intermediates covered in the assessment is somewhat smaller than that compiled by Khan et al. [Environ. Sci.: Processes Impacts 20, 437 (2018)]. The authors may want to comment explicitly on why they have focused on the more limited set of reactions.

Response: The referee is correct that, although we clearly state (e.g. in Sect. 1 and Fig. 2) which sCIs are considered in the evaluation, the reaction coverage is less well explained and justified. In Sect. 1, the following text (slightly adjusted because of the recent data for *E*-(CH=CH$_2$)(CH$_3$)COO – see comment A1) provides a broad indication of the scope of the evaluation relating to the reactions of sCIs:

> "The spectroscopy and kinetics recommendations for the sCI reactions are presented and discussed in Sects. 5 and 6. These include data for bimolecular and unimolecular reactions of selected sCIs of particular atmospheric relevance for which direct kinetic data have been reported, namely $CH_2OO$, *Z*- and *E*- $CH_3CHOO$, $(CH_3)_2COO$ and *E*-(CH=CH$_2$)(CH$_3$)COO; and we also provide some discussion of the complete set of C$_4$ intermediates formed from isoprene (see Fig. 2 for sCI structures)"

In Sect. 6.2 (*Evaluation of rate coefficients for bimolecular reactions*), we have now added the following explicit information on reaction coverage (new text in red font):

> "Table 5 provides a summary of the preferred values of bimolecular reaction rate coefficients, with additional details given in the corresponding reaction data sheets in Supplement B. As indicated in Sect. 1, the evaluation focuses on classes of reaction that are of particular significance in tropospheric chemistry. Where data are available, these include reactions with $SO_2$, $NO_2$, $H_2O$, $(H_2O)_2$, $CH_3CHO$, $CH_3C(O)CH_3$, $CF_3C(O)CF_3$, $HC(O)OH$, $CH_3C(O)OH$ and

The referee is correct that there are some reaction classes that are included in the tabulations of Khan et al. (2018) which we do not consider, these being reactions that are uncompetitive under tropospheric conditions; and we are grateful for the opportunity of clarifying this in the revised manuscript.

It should be noted, however, that our evaluation also includes some additional reactions and information not provided by Khan et al. (2018). These include speciated recommendations for, and discussion of, the complete set of four isoprene-derived C$_4$ sCIs for unimolecular decomposition and bimolecular reaction with SO$_2$, H$_2$O and (H$_2$O) (i.e. 16 reactions), for which Khan et al. (2018) only tabulate a single bulk parameter (not significantly different from zero) for unimolecular decomposition of the C$_4$ set collectively (from Newland et al. (2015)). In addition, we evaluate the data and provide a recommended rate coefficient for each reaction considered. In contrast, Khan et al. (2018) tabulate most laboratory and some theoretical determinations of the rate coefficients to inform their review and discussion of the atmospheric chemistry of sCIs, but provide no recommendations or preferred values.

**C. Additional changes**

The chemistry of sCIs is a fast-moving field, and a number of new studies have been reported during the open discussion of the paper. Some of these have been raised by the referees, and have been dealt with as described above. In addition, we have responded to publication of other studies by implementing the following changes and additions:

(i) **Yields of stabilised Criegee intermediates**: Newland et al. (2020) recently reported a systematic study of total sCI yields, and speciated yields of stabilised CH$_2$OO and (CH$_3$)$_2$COO, from the reactions of O$_3$ with a series of alkenes. These have resulted in small changes to the recommended total sCI yields for propene and 2-methypropene in Table 2, and a new entry for β-myrcene. This also alerted us to a prior study of Newland et al. (2018), that had been overlooked, resulting in further small changes to the recommended total sCI yields for β-pinene and limonene. The underlying information is also summarised and discussed in the relevant O$_3$ + alkene data sheets, which have been revised accordingly.

(ii) **Kinetics of CH$_2$OO decomposition and its reaction with HC(O)OH**: Peltola et al. (2020) have recently reported a kinetics study of CH$_2$OO. This was produced using a new source, namely the photolysis of CH$_2$IBr. As a result, the following change has been made to the text in Sect. 5.2 (*UV spectra of stabilised Criegee intermediates*):

"In experimental studies of UV-visible spectra, the series of C$_1$ – C$_3$ sCIs have mainly been formed by photolysis of the corresponding di-iodoalkane (via C-I bond fission), followed by the reaction of the iodoalkyl radical with O$_2$, e.g. in the case of CH$_2$OO (e.g. Beames et al., 2012; Sheps, 2013):

CH$_2$I$_2$ + hυ → CH$_2$I + I        (R11)

CH$_2$I + O$_2$ → CH$_2$OO + I        (R12)

The formation of CH$_2$I (and subsequently CH$_2$OO) from the photolysis of CH$_2$IBr (via C-Br bond fission) has also recently been reported (Peltola et al., 2020), suggesting that bromo-iodoakanes may also be used more widely as sCI precursors."

The reported kinetic data for CH$_2$OO decomposition and its reaction with HC(O)OH have been included in data sheets CGI_12 and CGI_11. This has resulted in changes to the recommended rate coefficients, as shown in Table 5 of the revised manuscript. In the case of CH$_2$OO decomposition (CGI_12), the recommended rate coefficient has been significantly simplified to an upper limit value at 298 K and 1 bar only. This is also discussed in Sect 7 (*Overall reactivity conclusions – comparison of experiment and theory*), where the value is now compared with that calculated for the same conditions by Long et al. (2016). However, this revision does not change the main conclusion that CH$_2$OO decomposition is an unimportant loss process, compared with other routes, under tropospheric conditions

[revised manuscript text omitted]

**Comments**

[revised manuscript text omitted]